# UCB Provably Learns From Inconsistent Human Feedback

Shuo Yang [1]   Tongzheng Ren [1]   Inderjit S. Dhillon [1]   Sujay Sanghavi [2]

## Abstract

In this paper, we study how to learn from inconsistent human feedback in the setting of combinatorial bandits with semi-bandit feedback – where an online learner in every time step chooses a size-$k$ set of arms, observes a stochastic reward for each arm, and endeavors to maximize the sum of the per-arm rewards in the set. We consider the challenging setting where these per-arm rewards are not only set-dependent, but also *inconsistent:* the expected reward of arm "a" can be larger than arm "b" in one set, but smaller in another. Inconsistency is often observed in practice, falls outside the purview of many popular semi-bandit models, and in general can result in it being combinatorially hard to find the optimal set.

Motivated by the observed practice of using UCB-based algorithms even in settings where they are not strictly justified, our main contribution is to present a simple assumption - weak optimal set consistency. We show that this assumption allows for inconsistent set-dependent arm rewards, and also subsumes many widely used models for semi-bandit feedback. Most importantly, we show that it ensures that a simple UCB-based algorithm finds the optimal set, and achieves $O\left(\min(\frac{k^3 n \log T}{\epsilon}, k^2 \sqrt{nT \log T})\right)$ regret (which nearly matches the lower bound).

## 1. Introduction

Combinatorial bandits (see e.g. (Chen et al., 2013; Saha & Gopalan, 2019)) model online learning settings where at each time a set of items has to be selected from a pool, and the learner subsequently observes *set-dependent rewards* which need to be incorporated into future set selection decisions. In this paper we are interested in stochastic combi-

natorial bandits with semi-bandit feedback (Combes et al., 2015) - that is, at each time we get to observe a stochastic reward for each arm in the set that was played, but the distribution of this observed arm reward depends on the set - and can be different when different sets containing the same arm. The reward of a set is the summation of the per-arm rewards in the set. Combinatorial bandits with semi-bandit feedback model many real-world sequential recommendation applications; take for example a setting where a slate of online advertisements have to be shown to users, who may subsequently either click on one of them or none of them. Here each ad corresponds to an arm, a set of ads (i.e. arms) needs to be selected and shown, and one of them would get a reward of "1" because it is clicked and the others would get a reward of "0". Combinatorial bandit settings allow for the click-probability of an ad (i.e. the value of an arm) to depend on the set in which it was presented (Saha & Gopalan, 2019).

In this paper, we are interested in **inconsistent and possibly contradictory preferences:** that is, we can have two arms $a$ and $b$ and two sets $s_1$ and $s_2$, so that in expectation, arm $a$ is more valuable than arm $b$ in set $s_1$, but $b$ is more valuable than $a$ in $s_2$. It has been repeatedly observed that human preferences are typically constructed only when offered a set of alternatives, and the preference can be inconsistent across different sets. For instance, an empirical study on "selection of college applicants" showed that evaluators often prefer student $A$ over $B$, $B$ over $C$ and $C$ over $A$ when compared in pairs of two (Tversky, 1969); people exhibit inconsistent preferences for eco-friendly products under different contexts (MacDonald et al., 2009); and the well-known "framing effect", where "reversals of preference are induced by changes in the reference points" (Kagel & Roth, 2020).

However, many popular parametric models (e.g. multinomial logits, random utility, etc.) do not allow for such inconsistencies in arm rewards. Indeed, in the worst case, general inconsistent preferences imply that the play of one set will reveal absolutely nothing about any other - an impossible situation for an online learner. Meanwhile, in practice, a simple approach is seen to work even with inconsistent preferences (e.g. the closely-related Sparring algorithm by Ailon et al., 2014): maintain a per-arm UCB as would be done if this was a simple non-combinatorial bandit problem,

[1]Department of Computer Science, University of Texas at Austin, TX, US [2]Department of Electrical and Computer Engineering, University of Texas at Austin, TX, US. Correspondence to: Shuo Yang <yangshuo_ut@utexas.edu>.

Interactive Learning with Implicit Human Feedback Workshop at ICML 2023.

| Algorithm | Regret | Best Set | Set-Dep. Reward | Inconsistent Preferences |
|---|---|---|---|---|
| CUCB (Chen et al., 2013) | $O\left(\frac{k^2 n \log T}{\epsilon}\right)$ | ✓ | ✗ | ✗ |
| CombUCB1 (Kveton et al., 2015b) | $O\left(\frac{kn \log T}{\epsilon}\right)$ | ✓ | ✗ | ✗ |
| ESCB (Combes et al., 2015) | $O\left(\frac{\sqrt{k}n \log T}{\epsilon}\right)$ | ✓ | ✗ | ✗ |
| MNL-TS (Agrawal et al., 2017) | $O\left(\sqrt{NT}\log TK\right)$ | ✓ | ✓ (MNL) | ✗ |
| Explor.-Exploit. (Agrawal et al., 2019) | $O\left(\frac{kn \log T}{\epsilon}\right)$ | ✓ | ✓ (MNL) | ✗ |
| MaxMin-UCB (Saha & Gopalan, 2019) | $O\left(\frac{n \log T}{\epsilon}\right)$ | ✗ | ✓ (MNL) | ✗ |
| Rec-MaxMin-UCB (Saha & Gopalan, 2019) | $O\left(\frac{n \log T}{k\epsilon}\right)$ | ✓ | ✓ (MNL) | ✗ |
| Choice Bandits (Agarwal et al., 2020a) | $O\left(\frac{n^2 \log n}{\epsilon^2} + \frac{n \log T}{\epsilon^2}\right)$ | ✗ | ✓ | ✓ |
| Algorithm 1 (Ours) | $O\left(\min\left(\frac{k^3 n \log T}{\epsilon}, k^2\sqrt{nT \log T}\right)\right)$ | ✓ | ✓ | ✓ |

*Table 1.* Regret upper bounds and settings for stochastic combinatorial bandits. The checkmarks in "Best Set" mean that the algorithms are designed for finding the best size-$k$ set, while cross marks mean the algorithms aim to identify the best arm. The checkmarks in "Set-Dep. Reward" represent that the reward distribution of arms depends on the set they reside in, while cross marks mean the rewards of the arms are generated independent of the set. The cross marks in "Inconsistent Preference" correspond to assuming individual arms to have intrinsic value, and therefore a consistent preference among the arms, while checkmarks are for the algorithms that do not require such assumption.

and in every step pick the $k$ arms for which these UCB estimates are the highest. This behavior is seen even though this is clearly a setting UCB was not designed for, and for which there is little theoretical understanding. We, therefore, seek a theoretical understanding of the following question:

*Why does UCB work for combinatorial bandits, even when the relative goodness of arms is inconsistent (and possibly contradictory) in different sets?*

Notice that **existing analysis of UCB does not provide a regret bound when the preference is set-dependent and inconsistent**, as there are no fixed expected rewards (or any notion of intrinsic value) associated with the arms. One suboptimal arm, under inconsistent preferences, can have large reward expectations in many suboptimal sets and therefore has large UCB, which potentially leads to linear regret.

In this paper, we present a surprisingly weak assumption that allows for inconsistent preferences, generalizes many popular parametric (and consistent) models, and most importantly guarantees that UCB finds the best set. In particular, our **main contributions** are summarized below:

- We first present the *weak optimal set consistency (WOSC)* assumption (Assumption 1). WOSC only requires that the reward for any individual arm in the optimal set should be less than the reward for that same arm when it is in any other set, and nothing else. This clearly allows for inconsistent orderings, but interestingly also includes as special cases many commonly adopted parametric reward models (e.g., multinomial logit, and random utility model, etc.).

- We next present a novel analysis of the UCB-based algorithm under the WOSC assumption. Here by UCB-based algorithm we refer to the method where (a) an upper confidence bound is maintained and updated for each arm in the classical way (i.e. ignoring that arm rewards are in reality set dependent), and (b) at each time we choose the $k$ arms with the highest upper confidence bound values; the algorithm is described formally in (Algorithm 1). We prove that this algorithm has a gap-dependent $O(nk^3 \log T/\epsilon)$ regret upper bound, as well as a gap-independent $O(k^2\sqrt{nT \log T})$ regret upper bound (Theorem 3). Here $n$ is the total number of arms, $k$ is the size of selected set $s$, $T$ is the time horizon and $\epsilon$ is the minimum gap between the optimal and sub-optimal set.

- Finally, we prove a regret lower bound $\Omega(\frac{n \log T}{\epsilon})$ (Theorem 4) under WOSC. The lower bound nearly matches the regret bound of UCB-based algorithm for constant $k$ (which is common in practice, e.g., constant number of advertisement displaying slots), up to logarithmic factors.

## 2. Related Work

**Established Reward Model with Consistent Preferences.** There is a large volume of reward models established for combinatorial bandits. Most of them rely on an (implicit) assumption of consistent preferences. The simplest reward model assumes the reward of each arm is generated independent of the selected set (Chen et al., 2013; Kveton et al., 2015b; Combes et al., 2015; Simchowitz et al., 2016). (Farias et al., 2013) assumes a non-parametric reward model, but still requires a consistent preference over the arms. Other works adopt more complicated models to capture

the set-dependent reward distribution and assume consistent preferences over the arms. For example, the Multinomial Logit Model (MNL) assumes a deterministic utility associated with each arm, which induces a consistent preference (Abeliuk et al., 2016; Agrawal et al., 2019; Saha & Gopalan, 2019; Flores et al., 2019). Désir et al. (2015); Blanchet et al. (2016) approximate the user's choice as a random walk on a Markov chain. Berbeglia (2016) shows that the discrete choice model and the Markov chain model can be viewed as instances of a "random utility model" (RUM), which also assumes a consistent preference.

As will be discussed in Section 3.3, all the mentioned reward models that assume consistent preferences are subsumed by our general "weak optimal set consistency" assumption (Assumption 1), and therefore are covered by our proposed algorithm and analysis.

**Inconsistent Preference.** There are also other works considering inconsistent preferences, but focus on problem settings different from ours. The Choice Bandits (Agarwal et al., 2020a) assumes there exists a single best arm that has the largest expected reward in any set. Further, many recent dueling bandits works (Ramamohan et al., 2016; Wu & Liu, 2016; Sui et al., 2018) also extend to inconsistent preferences (in the absence of Condorcet winner), and consider Copeland winner (Zoghi et al., 2015; Komiyama et al., 2016), Borda winner (Urvoy et al., 2013; Jamieson et al., 2015), and von Neumann winner (Dudík et al., 2015). Their goal, however, is to find the best single arm, instead of finding the optimal set of arms. Dimakopoulou et al. (2019) considers a setting where the preferences may not be consistent, but provides no theoretical regret guarantee. (Kale et al., 2010; Han et al., 2021) present results on adversarial combinatorial bandits, whereas our focus is on stochastic bandits problem.

**The Scope of Our Paper.** While our assumption is general and subsumes many previously studied combinatorial bandits settings as special cases, there are also important (and seemingly relevant) problems that do not fit into the scope of this paper. Here we distinguish our setting from several other widely studied settings.

**(a)** Dueling bandits (Yue et al., 2012) focuses on how one can learn from the comparisons of 2 arms (Zoghi et al., 2013; Komiyama et al., 2015) or multiple arms (Brost et al., 2016; Sui et al., 2017; Saha & Gopalan, 2018; 2019) to find the best arm. They all focus on recovering the single best arm, instead of the best set, which is different from our setting. **(b)** Bandits with submodular reward function (Streeter & Golovin, 2007; Streeter et al., 2010; Yue & Guestrin, 2011; Hazan & Kale, 2012; Gabillon et al., 2013; Chen et al., 2017) assumes that a arm's marginal contribution to the set reward is inversely proportional to the reward of a set.

Our "weak optimal set consistency" assumption is conceptually related to submodularity. However, our result is not directly comparable with the results for submodularity as the formal definition of submodularity involves sets with different sizes whereas we focus on fixed set size $k$. **(c)** Another seemingly relevant line of work is bandits with cascade feedback (Radlinski et al., 2008; Kveton et al., 2015a; Zong et al., 2016; Lagrée et al., 2016; Cheung et al., 2019), for which the reward depends on the position of the arm in the offered list, whereas ours consider the played set to be orderless. **(d)** Non-additive reward function. (Rhuggenaath et al., 2020; Agarwal et al., 2020b; 2021) focus on a setting where the reward of each individual arm is independently generated and the set reward is some non-additive function. Our focus, on the other hand, is the setting where the individual arms' rewards are set-dependent and the relative order is not consistent in different sets.

## 3. Problem Setup and WOSC Assumption

In this section, we first formally describe our combinatorial bandits with semi-bandit feedback problem, and then present a motivating example for inconsistent preferences. We then formally present the *weak optimal set consistency* (WOSC) assumption (Assumption 1) and show how it allows for inconsistent preferences. Further, we define the "consistent preferences" (Definition 1), and show that many widely studied models (MNL, RUM, etc.) assume consistent preferences and are covered by WOSC.

**Stochastic combinatorial multi-armed bandits problem with semi-bandit feedback.** Given a fixed set of arms $\mathcal{A} = \{a_1, a_2, \cdots, a_n\}$, let $\mathcal{S}$ denote all the size-$k$ subsets of $\mathcal{A}$. At each time step $t$, the online learner selects a set $s(t) \in \mathcal{S}$, and then observes a per-arm stochastic reward $X_{a,s(t)}$ of all its arms $a \in s(t)$. The stochastic reward for the set $s(t)$ is $\sum_{a \in s(t)} X_{a,s(t)}$ – i.e. the *set reward is the sum of the (set-dependent) per-arm rewards.*

This setup models, for example, click-through rates: every time a set is presented to a user, the learner gets to observe the arms which were clicked and which were not, and in turn is trying to maximize the total overall number of clicks. We allow the probability of an arm being clicked to depend on the set, and be possibly inconsistent across sets.

We denote the expected reward of arm $a$ in set $s$ to be $Q_s(a) \triangleq \mathbb{E}[X_{a,s}]$. The optimal set is denoted by $s^* \triangleq \arg\max_s \sum_{a \in s} Q_s(a)$, and finally, the regret for time $t$ is

$$reg(t) = \sum_{a \in s^*} Q_{s^*}(a) - \sum_{a \in s(t)} Q_{s(t)}(a),$$

and the regret up to time $T$ is $R(T) \triangleq \mathbb{E}\left(\sum_{t=1}^{T} reg(t)\right)$. The online learner aims to minimize $R(T)$.

Before formally introducing our assumption for inconsistent preferences, we first present a motivating example.

### 3.1. A Motivating Example

Consider a synthetic example of providing recommendations to a customer looking for cameras. There are 6 candidates {`Nikon, Sony, Canon, Digital Camera, Keyboard, Shoes`}. Every time we need to offer 3 recommendations and the customer accepts at most one of them. A customer's acceptance gives the recommendation reward 1 and otherwise gives reward 0.

Suppose the user is interested in {`Nikon, Sony, Canon`}, but when some of them are not recommended, the `Digital Camera` recommendation will partially capture the corresponding interest.

Specifically, we set the expected reward to be $Q.(\text{Nikon}) = 0.35, Q.(\text{Canon}) = 0.3, Q.(\text{Sony}) = 0.25, Q.(\text{Keyboard}) = 0.01, Q.(\text{Shoes}) = 0.01$, where the dot means any set containing the concerning recommendation. Further, set $Q_s(\text{Digital Camera}) = 0.85 - \sum_{a \in s, a \neq \text{Digital Camera}} Q_s(a)$. We show 4 representative sets in Figure 1. It can be verified that the optimal set is {`Nikon, Sony, Canon`} as the total expected reward is the highest 0.9.

Notice that the existence of the recommendation `Digital Camera` makes the problem harder. As shown in Figure 1, the `Digital Camera` has the highest accepting probability in many sets. Further, observe that a user is more likely to accept `Digital Camera` than `Nikon` in Set #3, whereas `Nikon` belongs to the optimal set. This makes `Digital Camera` seemingly a good recommendation, but it is not part of the optimal set.

Generally, it is conceivable that there are cases where the optimal arms are not the best in all sets, and there exist suboptimal arms that have higher expected rewards in many sets. The UCB-based algorithm (which is widely used as a heuristic under inconsistent preferences) can, therefore, over-value some suboptimal arms and thus has linear regret.

In the following sections, however, we theoretically show that the UCB-based algorithm has near-optimal regret even under some inconsistent preferences. The crux is adopting the *weak optimal set consistency* assumption, which is formally defined in the next subsection. The WOSC assumption allows for inconsistent preferences, subsumes many previously studied reward models, and most importantly, guarantees that UCB finds the optimal set.

### 3.2. WOSC Assumption

Here we formally define the WOSC assumption and show how it allows for inconsistent preferences.

**Assumption 1** (**Weak Optimal Set Consistency (WOSC)**). *Let $s^*$ be the optimal set, $s$ be any other set and arm $a \in s \cap s^*$ be in both sets. Then WOSC requires that $Q_s(a) \geq Q_{s^*}(a)$ – i.e. arm $a$'s individual reward in $s$ is larger than its individual reward in $s^*$.*

*Remark 1:* WOSC means that the optimal set $s^*$ is the most *competitive* set – while the *overall sum* of arm rewards is highest in $s^*$, any arm would have fared better in a different set because that set would have less competitive options.

*Remark 2:* One salient feature of WOSC is **not** assuming consistent preferences over arms $a \in \mathcal{A}$ at any time $t$. We first present an example that is allowed by our assumption but not other commonly seen reward models. We then formally discuss the "consistent preference" in the next subsection.

**Example 1.** *For any $k > 2$, without loss of generality, we take $a_1 \in s^*, a_2 \in s^*$ with $Q_{s^*}(a_1) \geq Q_{s^*}(a_2)$. For some sub-optimal set $s_i$, Assumption 1 allows for:*

1. ***Reversed relative reward expectation***:

   $$Q_{s^*}(a_1) \geq Q_{s^*}(a_2), \quad Q_{s_1}(a_2) > Q_{s_1}(a_1),$$
   *for some $s_1$ containing $a_1, a_2$.*

2. ***Non-transitive relative reward expectation***: *for some $s_4$ containing $a_2, a_3$, and $s_5$ containing $a_1, a_3$,*

   $$Q_{s^*}(a_1) > Q_{s^*}(a_2), \quad Q_{s_4}(a_2) > Q_{s_4}(a_3),$$
   $$Q_{s_5}(a_3) > Q_{s_5}(a_1).$$

Note that the $s_5$ in the "non-transitive" part of Example 1 also shows that Assumption 1 allows the arms not in $s^*$ to be better than the arms belonging to $s^*$ in some sub-optimal set. This corresponds to the `Digital Camera` in the motivating example in Section 3.1.

### 3.3. Existing Models are Strongly Consistent

Informally, consistent preferences mean that one arm is intrinsically more valuable than another, irrespective of the sets in which both those arms are presented. In this section, we first formally define consistent preferences, and then show that three widely used models – multinomial logit, random utility, and independent reward – implicitly assume consistent preferences. As mentioned above, our WOSC assumption covers cases that are not consistent; however, in this section, we show that it *also* covers any strongly consistent setting.

**Definition 1** (Strong Consistent Preferences). *Set dependent arm rewards $\{Q_s(a)\}$ are said to represent strong consistent preferences if there exists a **total ordering** of arms. In particular, for any pair of arms $a_i$ and $a_j$ such that $a_i \succ a_j$,*

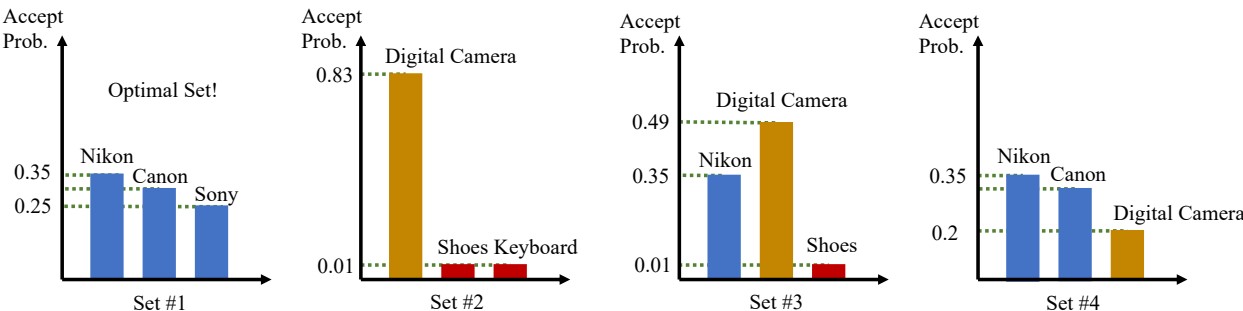

*Figure 1.* Four representative sets. The set #1 is optimal, as it maximizes the sum of the accepting probability of the recommendations. The `Digital Camera` has the highest accepting probability in many sub-optimal sets (even when paired with the recommendations belonging to the optimal set. See set #3). Such instances break the consistent preferences, but are covered by Assumption 1.

*and any corresponding pair of size-$k$ sets $s$ and $s'$ which differ only in these arms – i.e. $s = s' - a_j + a_i$ we have that the set-dependent arm rewards satisfy*

$$Q_s(a_i) \geq Q_{s'}(a_j) \text{ and } Q_s(a) \leq Q_{s'}(a), \; \forall a \in s \cap s'.$$

*Further, the total set rewards also satisfy $\sum_{a \in s} Q_s(a) \geq \sum_{a \in s'} Q_{s'}(a)$.*

We now show here that three widely adopted reward models all assume a "strong consistent preference", which are all also covered by WOSC (Assumption 1). For clarity of exposition, here we focus on the binary reward with $X_{a,s} \in \{0, 1\}$ and $Q_s(a)$ is therefore the probability that $a$ receives reward 1 in set $s$.

**Multinomial Logit (MNL):** MNL assumes a deterministic utility $v_i$ associated with each $a_i$ and the probability of $a_i$ receiving non-zero reward in $s$ is $Q_s(a_i) = \frac{e^{v_i}}{e^{v_0} + \sum_{a_j \in s} e^{v_j}}$, where $v_0$ is some constant modeling the event of no arm receiving non-zero reward. One can verify that the $v_i$s of MNL induce strong consistent preferences and the optimal set $s^*$ is composed by arms with highest $v_i$. Assumption 1 covers MNL since $e^{v_0} + \sum_{a_j \in s} e^{v_j} \leq e^{v_0} + \sum_{a_j \in s^*} e^{v_j}, \forall s \neq s^*$.

**Random utility model (RUM):** RUM assumes a (random) utility associated for all $a_i \in \mathcal{A}$, with $U_i = v_i + \epsilon_i$, where $v_i$ is a deterministic utility and $\epsilon_i$s are i.i.d. random variables drawn at every time step $t$. The probability of $a_i$ in $s$ receiving non-zero reward is given by $Q_s(a_i) = P(U_i > U_j, \forall a_j \in s \text{ and } i \neq j)$. To model the event of no arm $a \in s$ receiving non-zero reward, $s$ can be augmented to $s \cup \{a_0\}$, with random utility $U_0$ of $a_0$ defined similarly. When $U_0$ is the largest, no arm $a \in s$ receives non-zero reward. It can be verified that $v_i$s in RUM induce a strong consistent preference, and the optimal set $s^*$ is composed by arms with highest $v_i$. For any arm $a \in s^*$, putting it to a sub-optimal set $s$ leads to arm $a$ having a larger chance of receiving non-zero reward, as other arms have smaller $v_i$, thus satisfies Assumption 1.

**Independent reward:** Independent reward model assumes a deterministic reward expectation $v_i$ associated with arm $a_i$. For the arm $a_i$ in any set $s$, it assumes $Q_s(a_i) = v_i$. The $v_i$s immediately induce a strong consistent preference. The independent reward model is also covered by Assumption 1, as $Q_s(a_i)$ does not change in different $s$.

Finally, we show that the WOSC assumption represents a strict generalization of strong consistent preferences. That is, it subsumes all strongly consistent reward models, but also allows for inconsistent models.

**Lemma 2.** *Any reward model $\{Q_s(a)\}$ that represents strong consistent preferences also satisfies WOSC (Assumption 1). However, the reverse is not true; there exist reward models that satisfy WOSC but do not represent strong consistent preferences.*

We defer the proof to Appendix B.

## 4. UCB-based Algorithm and Regret Analysis

In this section, we formally describe the simple UCB-based algorithm for combinatorial bandits, and present its regret bounds (both gap-dependent and gap-independent). We present a novel way of analyzing UCB, while allowing set-dependent arm rewards as long as WOSC is satisfied.

We emphasize that our main contribution is not in algorithmic innovation, but in rigorously proving that the UCB-based algorithm achieves near-optimal regret under inconsistent preferences (Assumption 1).

### 4.1. Algorithm

Denote $N_i(t)$ to be the number of times that $a_i$ is included in the selected set $s$ up to time $t$, $C_i(t)$ to be the cumulative reward of arm $a_i$ at time $t$. We have Algorithm 1 that extends the standard $\alpha$-UCB algorithm. It selects a set of arms with top-$k$ UCB in each step. It is worth noting that Algorithm 1

only keeps track of the cumulative reward of the arms in $\mathcal{A}$, without accounting for any set-dependent information. Though it may seem contradictory to the set-dependent reward distribution, we will show that Algorithm 1 achieves near-optimal regret.

---

**Algorithm 1** UCB-BASED ALGORITHM FOR COMBINATORIAL BANDITS WITH INCONSISTENT PREFERENCES

---

1: **Online learning task:** Given a set of arms, in each time choose a size-$k$ subset so as to maximize reward. Feedback in every step is a stochastic reward for every arm in that chosen set; the distribution of these rewards can be set-dependent, and inconsistent across sets.
2: **Input:** arm set $\mathcal{A}$ of size $n$, set size $k$, time horizon $T$, rewards bounded by $B$
3: **Parameter:** A constant $\alpha$, normally set to 2
4: **Initialize:** $UCB_i(1) = INF$, $N_i(1) = 0$, $C_i(1) = 0$ for all arm $a_i \in \mathcal{A}$
5: **for** $t = 1$ **to** $T$ **do**
6:    Construct set $s(t)$ with arms that have top-$k$ $UCB_i(t)$, ties break randomly. For all $a_i \in s(t)$, Set $N_i(t+1) = N_i(t) + 1$
7:    Observe feedback. Set $C_i(t+1) = C_i(t) + X_{a_i,s(t)}$
8:    $UCB_i(t+1) = \frac{C_i(t+1)}{N_i(t+1)} + B\sqrt{\frac{\alpha \log T}{N_i(t+1)}}$, for all arm $a_i \in s(t)$, and $UCB_i(t+1) = UCB_i(t)$, for others
9: **end for**

---

### 4.2. Regret Bound

Let $\epsilon = \sum_{a \in s^*} Q_{s^*}(a) - \max_{s \neq s^*} \sum_{a \in s} Q_s(a)$ denote the minimum gap in expected reward between the optimal set $s^*$ and any sub-optimal set $s$. Recall that $k$ is the size of the selected set $s$, and $n$ is the size of $\mathcal{A}$. Suppose the reward $X_{a,s}$ is bounded by $B$ (i.e., $X_{a,s} \in [0, B]$) for all $a$ and $s$. Our next result provides a regret bound of Algorithm 1.

**Theorem 3** (Regret Bound of Algorithm 1)**.** *For combinatorial bandits problem under Assumption 1, run Algorithm 1 with parameter $\alpha \geq 2$, we have*

$$R(T) \leq O\left(\min\left(\frac{B^2 k^3 n \log T}{\epsilon}, Bk^2 \sqrt{nT \log T}\right)\right).$$

For the "well-separated" problem (i.e., $\epsilon$ is large), the regret scales with $\log T$; and when the sub-optimality gap $\epsilon$ is small, the gap-independent bound $Bk^2\sqrt{nT \log T}$ will dominate the $\min$, and this recovers the standard gap-independent $\widetilde{O}(\sqrt{T})$ regret scaling.

Further, we have the following regret lower bound for the combinatorial bandits under WOSC.

**Theorem 4** (Regret Lower Bound)**.** *For any online learning algorithm that achieves $o(T^c)$ regret for all constant $c > 0$, there exists a problem instance that satisfies Assumption 1, such that the algorithm induces a regret of $\Omega\left(\frac{B^2 n \log T}{\epsilon}\right)$.*

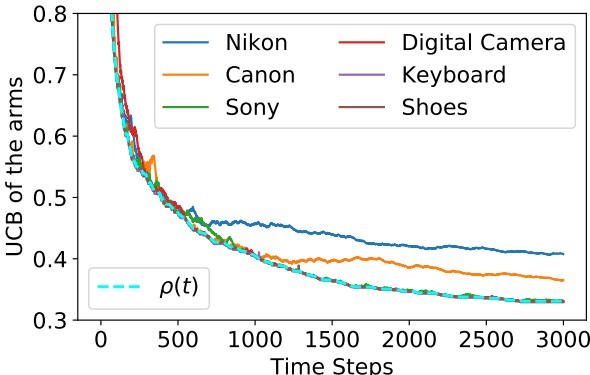

*Figure 2.* Evolution of UCB in the environment defined in Section 3.1. **Observation:** the UCB of all arms decreases together initially, and the arms in the optimal set separate out later. $\rho(t)$ (a "lower bound" of the played arms' UCB, formally defined Section 4.3) precisely captures this decreasing-together dynamics. As $\rho(t)$ decreases, the UCB of two recommendations `Nikon`, `Canon`, belonging to the optimal set, separate out from $\rho(t)$, and are therefore included in all the subsequently played sets. This happens with `Digital Camera` having the largest reward expectation in many suboptimal sets.

The dependency of $B, n, T, \epsilon$ in the lower bound matches the gap-dependent upper bound (Theorem 3). For $k$ being constant, which is commonly seen in practice (e.g., a constant number of displaying slots in online recommendation systems), our regret bound nearly matches the lower bound up to logarithmic terms. This shows the optimality of Algorithm 1 despite inconsistent preferences.

**Regret Upper Bound Analysis Intuition** To see how the UCB-based algorithm works under WOSC, we first present an illustrative experiment here. The environment is the motivating example presented in Section 3.1, where the optimal set is {`Nikon`, `Canon`, `Sony`} but `Digital Camera` has the highest reward expectation in many sets.

Figure 2 shows the process of the UCB-based algorithm converging to the optimal set. The observation is that the UCB of most arms decreases together, and the UCB of the optimal arms sequentially separates out.

The regret analysis follows this observation closely and can be summarized as 3 steps:

- *Step I:* proving that the UCB of the optimal arms stays large. The WOSC assumption is invoked here, which guarantees that the UCB of an optimal arm $a_i$ is always larger than $Q_{s^*}(a_i)$.

- *Step II:* showing that the UCB of most of the arms decreases together and stays close to each other (characterized by $\rho(t)$, see Figure 2 and definition in Section 4.3).

- *Step III:* showing $\rho(t)$ can not stay high for a long time, which is then converted into a regret bound.

All our analysis focuses on characterizing the dynamics of UCB – how the UCB of the optimal arms and other arms decays. It does not require the arms to have fixed reward expectations or consistent preferences, which is drastically different from the standard UCB analysis.

### 4.3. Proof Sketch

The following results are for Algorithm 1 with $\alpha \geq 2$. We sketch the proof into 3 steps, which correspond to our discussion for analysis intuition. W.l.o.g., let $s^* = \{a_1, a_2, \cdots, a_k\}$ with $Q_{s^*}(a_1) \geq \cdots \geq Q_{s^*}(a_k)$.

**Step I: UCB of optimal arms stays large.**

We first show that the $UCB_i(t)$ of $a_i \in s^*$ is lower bounded by $Q_{s^*}(a_i)$ for all time steps, with high probability.

**Lemma 5** (UCB is optimistic). *With probability at least $1 - \frac{2}{T}$, we have $UCB_i(t) \geq Q_{s^*}(a_i)$ simultaneously for all time step $t \in [T]$ and all arm $a_i \in s^*$.*

This follows from the WOSC assumption, where the expected reward $Q_s(a_i) \geq Q_{s^*}(a_i)$ for all $s \neq s^*$ and $a_i \in s \cap s^*$. Therefore, as long as $UCB_i(t)$ is an optimistic estimate (i.e., $UCB_i(t) \geq \sum_{\tau=1}^{t} Q_{s(\tau)}(a_i)/N_i(t)$, see Corollary 10), we have that $UCB_i(t) \geq Q_{s^*}(a_i)$, with high probability.

**Step II: UCB of most arms decay together as $\rho(t)$.**

Here we formalize the observation in Figure 2. Let $\rho'(t) = \min_{a_i \in s(t)} UCB_i(t)$, and $\rho(t) = \min_{\tau \leq t} \rho'(\tau)$. By definition, $\rho(t)$ is monotonically non-increasing, and $UCB_i(t) \geq \rho'(t) \geq \rho(t), \forall a_i \in s(t)$, (i.e., $\rho(t)$ is a lower bound for the UCB of the arms in $s(t)$).

The following lemma shows that for the arms not in $s(t)$, $\rho(t)$ is always an upper bound, and soon a tight estimate of all their UCB.

**Lemma 6** (Dynamics of UCB). $\rho(t) \geq UCB_i(t) \geq \rho(t)\left(1 - \frac{1}{N_i(t)}\right), \forall a_i \notin s(t), \forall t \in [T]$.

*Proof.* For any arm $a_i \notin s(t)$, let $t' \leq t$ be the last time step that $a_i \in s(t')$. We then have

$$C_i(t') + \sqrt{\alpha N_i(t') \log T} \geq \rho'(t')N_i(t')$$
$$\geq \rho(t')N_i(t') \geq \rho(t)N_i(t').$$

The last step holds as $\rho(t)$ is non-increasing. With $C_i(t) \geq C_i(t')$ and $N_i(t) = N_i(t') + 1$, we have

$$C_i(t) + \sqrt{\alpha N_i(t) \log T} \geq \rho(t)\left(N_i(t) - 1\right).$$

Dividing both sides by $N_i(t)$ gives the second inequality. It is left to show $\rho(t) \geq UCB_i(t), \forall a_i \notin s(t)$. Let $t'' \leq t$ be the last time step $\rho'(t'') = \rho(t)$. It implies

$$\rho'(\tau) > \rho'(t'') = \rho(t) \geq UCB_i(t''),$$
$$\forall \tau \in (t'', t], a_i \notin s(t'').$$

Notice that $UCB_i(\tau+1) = UCB(\tau)$ if $a_i \notin s(\tau)$. Therefore for any $a_i \notin s(t'')$, it implies $a_i \notin s(\tau), \forall \tau \in [t'', t]$.

Notice that there are $(n - k)$ arms not in $s(t)$ and the same number of arms not in $s(t'')$, we have $a_i \notin s(t'') \iff a_i \notin s(t)$. Thus

$$UCB_i(t) = UCB_i(t'') \leq \rho'(t'') = \rho(t), \forall a_i \notin s(t).$$

This completes the proof. $\square$

Note that Lemma 6 implies that all arms $a_i$ with $UCB_i(t) \geq \rho(t)$ are included in $s(t)$. Therefore, combining with Lemma 5, we know that for all $a_i \in s^*$, with probability at least $1 - \frac{2}{T}$, once $\rho(t)$ falls below $Q_{s^*}(a_i)$, the subsequently played sets $s(t)$ will always contain $a_i$. As $\rho(t)$ keeps decreasing, the optimal set $s^*$ will be recovered sequentially, from $a_1$ to $a_k$.

**Step III: $\rho(t)$ can not stay large for long.**

The rest of the proof focuses on characterizing how fast $\rho(t)$ decays and converting it to a regret bound. Notice that $\rho(t)$ is the rolling-min of $\rho'(t)$ and is, therefore, monotonically non-increasing by definition. For $l \leq k$, let time $t_l$ be the last time that $\rho(t_l) \geq Q_{s^*}(a_l)$. Let $t'_l$ be the number of times that $s^*$ is selected before $t_l$. Lemma 7 presents a bound for the number of times that sub-optimal sets are played before $t_l$, which is measured by $t_l - t'_l$.

**Lemma 7** (Bound the times of selecting sub-optimal set). *With probability at least $1 - \frac{2}{T}$, we can bound $t_l - t'_l$, for all $l \in [k]$ as,*

$$t_l - t'_l \leq \frac{40\alpha B^2 lkn \log T}{(\Delta_l + \epsilon)^2}, \text{ if } \Delta_l \geq \frac{\epsilon}{10}; \text{ and}$$

$$t_l - t'_l \leq \frac{40\alpha B^2 lkn \log T}{\epsilon^2}, \text{ otherwise,}$$

*where $\Delta_l := \sum_{i=l}^{k} [Q_{s^*}(a_l) - Q_{s^*}(a_i)]$.*

*Remark 3:* Suppose $\rho(T) \geq Q_{s^*}(a_l)$ for some $a_l \in s^*$ (i.e., $\rho(t)$ does not fall below $Q_{s^*}(a_l)$ for the entire time horizon), we have that $t_l = T$ by definition, and Lemma 7 still holds.

We emphasize that Lemma 7 is crucial to prove regret bound with inconsistent preferences (Assumption 1), as it does not rely on each arm having a set-independent reward expectation, which is drastically different from existing UCB analysis. The next lemma connects regret $R(T)$ to $t_l - t'_l$ for $l \leq k$.

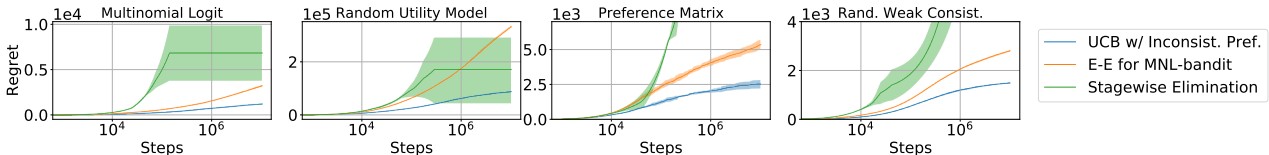

*Figure 3.* Synthetic experiments with different reward models. The curves are the averages and standard deviations of 5 independent runs. The "UCB w/ Inconsist. Pref." is Algorithm 1 with $\alpha = 2$. "E-E MNL-bandit" refers to the "Exploration-Exploitation algorithm for MNL-Bandit" (Agrawal et al., 2019). "Stagewise Elimination" was proposed in (Simchowitz et al., 2016). The parameters are specified as in the original papers.

**Lemma 8** (Regret decomposition)**.** *With probability at least* $1 - \frac{2}{T}$, *we have*

$$R(T) \leq 2B\sqrt{\alpha kn(t_k - t'_k)\log T}$$
$$+ \sum_{l=1}^{k-1} \delta_{lk}(t_l - t'_l) + nQ_{s^*}(a_k).$$

*where* $\delta_{ij} := Q_{s^*}(a_i) - Q_{s^*}(a_j)$.

Combining Lemmas 7 and 8 proves Theorem 3.

## 5. Experiments

We empirically evaluate Algorithm 1 on environments with different reward models (see Figure 3) satisfying the WOSC assumption. It demonstrates that the UCB-based algorithm is indeed able to deliver great empirical performance under WOSC, despite the set-dependent reward distributions and inconsistent preferences. We summarize the environments below, with details provided in Appendix E.

**Multinomial Logit**: We generate $n = 20$ arms, where each arm $a_i$ has an intrinsic value $v_i = \log(1 - 0.04i)$. The MNL model is used to determine the reward expectation $Q_s(a_i) = \frac{e^{v_i}}{e^{v_0} + \sum_{a_j \in s} e^{v_j}}$, with $v_0 = 0$. The set size is set to $k = 10$ and the number of possible sets is $184, 756$.

**Random Utility Model**: We generate $n = 20$ arms, where each arm $a_i$ has an intrinsic utility $v_i = 1 - 0.04i$. In every step, the random utility $U_i$ of all arms in the set $s(t)$ are independently generated with mean $\mu_i$ and unit variance from Gaussian distribution. Besides that, the null arm $a_0$ will draw a $U_0 \sim \mathcal{N}(2, 1)$. If $U_0$ is the maximum, then the entire set receives 0 reward. Otherwise, the arm with the largest random utility $U_i$ receives the reward 1 and others receive 0. The set size is set to $k = 5$ and the number of possible sets is $15, 504$.

**Preference Matrix**: We set the total number of arms to $n = 10$ and the set size to $k = 2$, then directly specify a 10-by-10 preference matrix $M$ to determine the probability of an arm receiving a reward. In particular, we set the matrix such that the preference is inconsistent – while $s^* = \{a_1, a_2\}$ with $Q_{s^*}(a_1) = 0.47$ and $Q_{s^*}(a_2) = 0.45$, the suboptimal arm

$a_3$ has a large reward expectation when paired with other suboptimal arms (i.e., $Q_s(a_3) = 0.675$ for $s$ not containing $a_1, a_2$). Therefore $a_3$ has the potential of being falsely recognized as an optimal arm, which leads to linear regret.

**Random Weak Optimal Set Consistency**: We randomly generate the environment that satisfies Assumption 1 via rejection sampling. We set the total number of arms to $n = 10$ and the set size to $k = 5$. Notice that these randomly generated environments need not satisfy the assumption of the MNL model (or RUM) other than Assumption 1.

Along with Algorithm 1, we also take "E-E for MNL-bandit" (Exploration-Exploitation algorithm for MNL, (Agrawal et al., 2019)) and "Stagewise Elimination" (Simchowitz et al., 2016) for comparisons, which are designed for "Multinomial Logit" and "Random Utility Model" environment, respectively. The algorithms are tested in the environments listed above. The average regret and standard deviation of 5 independent runs are reported in Figure 3.

"E-E for MNL-bandit" and "Stagewise Elim" perform relatively well in the environments that they are designed for. Note that in the "Preference Matrix" environment and "Random Weak Optimal Set Consistency" environment, there is no consistent preference among the arms. The "Stagewise Elimination" falsely eliminates an arm that belongs to the optimal set, and therefore suffers from linear regret. Despite the inconsistent preferences, all evaluated environments satisfy WOSC (Assumption 1) and the UCB algorithm (Algorithm 1) performs the best in all the testing environments.

## 6. Conclusion

In this paper, we study the combinatorial bandits with semi-bandit feedback under inconsistent preferences. We formally present a general assumption: weak optimal set consistency (WOSC), which allows for inconsistent preferences and subsumes many existing preference models (MNL, RUM, etc.). Under the WOSC assumption, we present a novel analysis for the UCB-based algorithm, which is widely used under inconsistent preferences as a heuristic algorithm with little theoretical understanding. Our analysis shows that, for constant $k$ (the set size), the simple UCB-based algorithm is nearly optimal under WOSC.

## Acknowledgement

This work is supported by NSF grant 1934932.

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

# A. Regret Lower Bound Proof

*Proof.* We prove the lower bound by constructing a family of environments $\mathcal{E}_i, i \in [n]$. We define the arm set as $\mathcal{A} = \{a_1, \cdots, a_{n+k-1}\}$ and consider the binary reward $X_{a_i, s} \in \{0, B\}$. In particular, we focus on the regime where $n \gg k$ and $B > 1$.

In environment $\mathcal{E}_i$, the optimal set is $\{a_i, a_{n+1}, a_{n+2} \cdots, a_{n+k-1}\}$. We assume the arms $\{a_1, a_{n+1}, a_{n+2} \cdots, a_{n+k-1}\}$ to have $\frac{1}{2}$ probability of receiving reward $B$ in any set in any environment, and arm $a_i$ has probability $\frac{1}{2} + \frac{\epsilon}{B}$ of receiving reward $B$ in any set in environment $\mathcal{E}_i$ for $i \in [2, n]$. All other arms not belonging to the optimal set have $\frac{1}{2} - \frac{\epsilon}{B}$ probability of receiving positive reward in any set. It's easy to verify that all environments $\mathcal{E}_i$ satisfies Assumption 1 and the minimum gap between optimal and sub-optimal set is $\epsilon$. We then have the following regret lower bound.

Let $q_i$ be the probability measure in environment $\mathcal{E}_i$. The proof follows by showing that, for all $j \in [2, n]$, any algorithm has $\mathbb{E}_{q_1}(N_j(T)) = \Omega(B^2 \log T/\epsilon^2)$ when the algorithm achieves $o(T^a)$ regret in environment $\mathcal{E}_j$ and $\mathcal{E}_1$.

For any $j \in [2, n]$, define the event $B_j = \{N_j(T) \leq B^2 \log T/\epsilon^2\}$. We prove the lower bound on $\mathbb{E}_{q_1}(N_j(T))$ by two cases. We first start with the simple one:

**Case I:** $q_1(B_j) < 1/3$. We have

$$\mathbb{E}_{q_1}(N_j(T)|q_1(B_j) < 1/3) \geq q_1(B_j^c) B^2 \log T/\epsilon^2 = \Omega(B^2 \log T/\epsilon^2).$$

**Case II:** $q_1(B_j) \geq 1/3$. Note that in environment $\mathcal{E}_j$, the algorithm will incur at least $\epsilon$ regret if not selecting $a_j$, Therefore we have $\mathbb{E}_{q_j}(T - N_j(T)) = o(T^c)$ for any constant $c > 0$. By Markov's inequality, we have

$$q_j(B_j) = q_j\left(\{T - N_j(T) > T - B^2 \log T/\epsilon^2\}\right) \leq \frac{\mathbb{E}_{q_j}(T - N_j(T))}{T - B^2 \log T/\epsilon^2} = o(T^{c-1}).$$

From (Karp & Kleinberg, 2007), we know that for any event $E$ and two distributions $p, q$ with $p(E) > 1/3$ and $q(E) < 1/3$, we have

$$D_{\mathrm{KL}}(p; q) \geq \frac{1}{3} \log(\frac{1}{3q(E)}) - \frac{1}{e},$$

where $D_{\mathrm{KL}}(p; q)$ is the KL-divergence of $p$ and $q$. Putting $q_1, q_j$ and $B_j$ into the inequality above, we have

$$D_{\mathrm{KL}}(q_1; q_j) \geq \frac{1}{3} \log(\frac{1}{3o(T^{c-1})}) - \frac{1}{e} = \Omega(\log T).$$

On the other hand, we need to bound the KL-divergence of $q_1$ and $q_j$ by playing any set containing $a_j$. Suppose $p$ is a categorical distribution with parameters $p_1, ..., p_k$ for $k$ items and $p'$ is another categorical distribution with parameters $p_1 - \epsilon_1, ..., p_k - \epsilon_k$, We have

$$D_{\mathrm{KL}}(p, p') = \sum_{i=1}^k (p_i' + \epsilon_i) \log \frac{p_i' + \epsilon_i}{p_i'} \leq \sum_{i=1}^k (p_i' + \epsilon_i) \frac{\epsilon_i}{p_i'} = \sum_{i=1}^k \frac{\epsilon_i^2}{p_i'},$$

where the last inequality holds because $\sum_{i=1}^k \epsilon_i = 0$. Since the only different arm between $\mathcal{E}_1$ and $\mathcal{E}_j$ is arm $a_j$ and the probability of $a_j$ receiving reward $B$ in environment $\mathcal{E}_j$ is $\frac{1}{2} + \frac{\epsilon}{B} \geq \frac{1}{3}$. We can directly bound the KL-divergence of $q_1$ and $q_j$ by

$$D_{\mathrm{KL}}(q_1; q_j) \leq 3N_j(T) \frac{\epsilon^2}{B^2},$$

It then directly implies that

$$3N_j(T) \frac{\epsilon^2}{B^2} = \Omega(\log T) \implies \mathbb{E}_{q_1}(N_j(T)|q_1(B_j) \geq 1/3) = \Omega\left(\frac{B^2 \log T}{\epsilon^2}\right).$$

Therefore, combining **Case I** and **Case II**, we have $\mathbb{E}_{q_1}(N_j(T)) = \Omega(B^2 \log T/\epsilon^2)$, which holds for all $j \in [2, n]$. Notice that playing $a_j$ induces $\epsilon$ regret, considering all $j \in [2, n]$ gives a regret lower bound $\Omega(B^2 n \log T/\epsilon)$. $\qquad \square$

# B. Proof for Section 3

## B.1. Proof of Lemma 2

*Proof.* We first show that any model assuming strong consistent preferences (Definition 1) is covered by WOSC (Assumption 1). By Definition 1, we have the optimal set $s^*$ is composed by the arms with largest $v_i$. For any sub-optimal set $s$, we can construct a sequence of sets $s_1, s_2, \cdots, s_m$, with $s_1 = s^*$, $s_m = s$ and each intermediate set $s_{i+1}$ changes one arm from $s_i$ into an arm from $s$. As we start from $s^*$ and going from $s_i$ to $s_{i+1}$, we always change an arm into some other arm with smaller $v$. Therefore for any arm $a \in s^* \cap s$, we have $Q_{s_{i+1}}(a) \geq Q_{s_i}(a)$. This implies that $Q_s(a) \geq Q_{s^*}(a)$, which matches Assumption 1.

On the other hand, a model assuming WOSC may not satisfy strong consistent preferences. Consider sets $s_1, s_2$ containing $a_i$ and $s_1', s_2'$ obtained by replacing $a_i$ by $a_j$, such that $Q_{s_1}(a_i) > Q_{s_1'}(a_j)$ and $Q_{s_2}(a_i) < Q_{s_2'}(a_j)$, the weak optimal set consistency (Assumption 1) allows for such case but it does not satisfy Definition 1.

This completes the proof. □

# C. Technical Lemmas

**Lemma 9.** *For $B$-bounded rewards $X_{a,s}$ (i.e., $X_{a,s} \in [0, B]$) and $\alpha \geq 2$, with probability at least $1 - \frac{2}{T}$, we have the following inequality holds for all arm $a_i$ and all time step $t \in [T]$ simultaneously:*

$$\left| C_i(t) - \sum_{\tau=1}^{t} Q_{s(\tau)}(a_i) \right| \leq B\sqrt{\alpha N_i(t) \log T}.$$

*Proof.* Recall that $N_i(t)$ is the number of times that arm $a_i$ is played up to time $t$. Let $\tau_j$ be the time step of the $j$-th pulling of arm $a_i$. Notice that if $a_i$ is not played at $t'$, then both $C_i(t') = C_i(t' - 1)$ and $Q_{s(t')}(a_i) = 0$. Therefore, we have that

$$C_i(t) - \sum_{\tau=1}^{t} Q_{s(\tau)}(a_i) = C_i(\tau_{N_i(t)}) - \sum_{j=1}^{N_i(t)} Q_{s(\tau_j)}(a_i).$$

Consider the quantity

$$D_i(q) = C_i(\tau_q) - \sum_{j=1}^{q} Q_{s(\tau_j)}(a_i),$$

where $q \in \{0, 1, \cdots, N_i(t)\}$. $D_i(0)$ to $D_i(N_i(t))$ is a martingale and $D_i(N_i(t)) = C_i(t) - \sum_{\tau=1}^{t} Q_{s(\tau)}(a_i)$. Consider the bad event $\mathcal{B}_{i,m}(t) := \left\{ N_i(t) = m \quad \text{and} \quad |D_i(N_i(t))| > B\sqrt{\alpha N_i(t) \log T} \right\}$, which can be interpreted as "the desired inequality fails at time step $t$ and the arm $a_i$ is played for $m$ times". By Azuma's inequality, we have

$$P(\mathcal{B}_{i,m}(t)) \leq 2 \exp\left( \frac{-2\alpha m B^2 \log T}{mB^2} \right) = \frac{2}{T^{2\alpha}}.$$

Consider event $\mathcal{B}_i(t) := \left\{ |D_i(N_t(t))| > B\sqrt{\alpha N_i(t) \log T} \right\}$, which can be interpreted as "the desired inequality fails at time step $t$ for arm $a_i$". We have $\mathcal{B}_i(t) = \cup_{m \in [t]} \mathcal{B}_{i,m}(t)$ and, therefore, with a union bound over $m \in [t]$,

$$P(\mathcal{B}_i(t)) \leq \frac{2t}{T^{2\alpha}} \leq \frac{2}{T^{2\alpha-1}}.$$

Further, with a union bound for all $i \in [n]$ and $t \in [T]$, we have that

$$P\left( \exists t \leq T, \exists i \in [n], \text{ s.t. } \left| C_i(t) - \sum_{\tau=1}^{t} Q_{s(\tau)}(a_i) \right| > B\sqrt{\alpha N_i(t) \log T} \right) \leq \frac{2n}{T^{2\alpha-2}}.$$

Notice that for $\alpha = 2$ and $T > n$, the inequality above implies that

$$\left| C_i(t) - \sum_{\tau=1}^{t} Q_{s(\tau)}(a_i) \right| \leq B\sqrt{\alpha N_i(t) \log T}.$$

holds simultaneously for all $i \in [n]$ and $t \in [T]$, with probability at least $1 - \frac{2}{T}$. $\qquad\square$

**Corollary 10** (UCB is optimistic). *For B-bounded rewards $X_{a,s}$ (i.e., $X_{a,s} \in [0, B]$) and $\alpha \geq 2$, with probability at least $1 - \frac{2}{T}$, we have the following inequality holds for all arm $a_i$ and all time step $t \in [T]$ simultaneously:*

$$UCB_i(t) \geq \frac{\sum_{\tau=1}^{t} Q_{s(\tau)}(a_i)}{N_i(t)}.$$

**Corollary 11** (Corollary of Lemma 9). *For B-bounded rewards $X_{a,s}$ (i.e., $X_{a,s} \in [0, B]$) and $\alpha \geq 2$, with probability at least $1 - \frac{2}{T}$, we have the following inequality holds for all arm $a_i$ and all time step $t \in [T]$ simultaneously:*

$$2B\sqrt{\alpha N_i(t) \log T} \geq N_i(t)UCB_i(t) - \sum_{\tau=1}^{t} Q_{s(\tau)}(a_i).$$

Without loss of generality, we assume $s^* = \{a_1, a_2, \cdots, a_k\}$ with $Q_{s^*}(a_1) \geq Q_{s^*}(a_2) \geq \cdots \geq Q_{s^*}(a_k)$. Recall that $\rho(t)$ is monotonically non-increasing by definition. Let time $t_l$ be the last time $t$ that we have $\rho(t) \geq Q_{s^*}(a_l)$ for $l \leq k$, we have the following result. Notice that if there exists an $l$ such that $\rho(T) \geq Q_{s^*}(a_l)$, we have $t_l = T$ by definition.

**Lemma 12.** *With probability at least $1 - \frac{2}{T}$, for all time steps $t > t_l$, we have $\{a_1, a_2, \cdots, a_l\} \subset s(t)$.*

*Proof.* The rest of the proof conditions on the event that the inequality in Lemma 9 holds, which happens with probability at least $1 - \frac{2}{T}$.

By definition of $UCB_i(t)$, we have

$$UCB_i(t) = \frac{C_i(t)}{N_i(t)} + \sqrt{\frac{\alpha \log T}{N_i(t)}} \geq \frac{\sum_{\tau=1}^{t} Q_{s(\tau)}(a_i)}{N_i(t)}, \quad \forall a_i \in s^*, \forall t \in [T]$$

Further, by WOSC, we have that $Q_s(a_i) \geq Q_{s^*}(a_i), \forall a_i \in s^*$ and all $s$ contains $a_i$. Therefore we have $\frac{\sum_{\tau=1}^{t} Q_{s(\tau)}(a_i)}{N_i(t)} \geq Q_{s^*}(a_i)$ and thus $UCB_i(t) \geq Q_{s^*}(a_i), \forall a_i \in s^*, \forall t \in [T]$.

Notice that Lemma 6 states that for all $a_i \notin s(t)$, we have $\rho(t) \geq UCB_i(t)$. It therefore implies $a_i \in s(t)$ if $UCB_i(t) > \rho(t)$. When $\rho(t) < Q_{s^*}(a_l)$ (which happens after $t_l$ by the definition of $t_l$ and the fact that $\rho(t)$ is non-increasing), we have $UCB_i(t) \geq Q_{s^*}(a_i) > \rho(t), \forall i \in [l]$. Therefore we have $a_i \in s(t), \forall i \in [l], \forall t > t_l$.

Note that when $t_l = T$, the Lemma 12 trivially holds true as there is no time step $t$ after $T$. Therefore, it concludes the proof that with probability at least $1 - \frac{2}{T}$, for all time steps $t > t_l$, we have $\{a_1, a_2, \cdots, a_l\} \subset s(t)$. $\qquad\square$

Let $\delta_{ij} \triangleq Q_{s^*}(a_i) - Q_{s^*}(a_j)$. Recall that $t_l$ is the last time step with $\rho(t_l) \geq Q_{s^*}(a_l)$ we have the following result.

**Lemma 13.** *With probability at least $1 - \frac{2}{T}$, we have the following results hold simultaneously for all $l \in [k]$:*

$$B\sqrt{4\alpha kn \left( t_l - \frac{l}{k} t'_l \right) \log T} \geq \sum_{i=1}^{l} Q_{s^*}(a_i) t_l + (k-l)Q_{s^*}(a_l)t_l - \sum_{i=1}^{n}\sum_{\tau=1}^{t_l} Q_{s(\tau)}(a_i) - \sum_{i=1}^{l-1} \delta_{il}(t_i - t'_i) - nQ_{s^*}(a_l),$$

*and*

$$B\sqrt{4\alpha kn \left( t_l - t'_l \right) \log T} \geq \sum_{i=k+1}^{n} Q_{s^*}(a_l) N_i(t_l) + \sum_{i=1}^{k} Q_{s^*}(a_i)N_i(t_l) - \sum_{i=1}^{n}\sum_{\tau=1}^{t_l} Q_{s(\tau)}(a_i) - nQ_{s^*}(a_l).$$

*Proof.* The rest of the proof conditions on the event that the inequality in Lemma 9 holds (and therefore the inequalities in Corollary 11 and lemma 12 hold), which happens with probability at least $1 - \frac{2}{T}$.

**Proof of the first inequality.** Recall that by Lemma 6, we have

$$UCB_i(t) \geq \rho(t) \left(1 - \frac{1}{N_i(t)}\right), \forall a_i \notin s(t),$$

and recall the definition of $\rho'(t) = \min_{a_i \in s(t)} UCB_i(t)$ and $\rho(t) = \min_{\tau \leq t} \rho'(\tau)$, we have

$$UCB_i(t) \geq \rho'(t) \geq \rho(t) \geq \rho(t) \left(1 - \frac{1}{N_i(t)}\right), \forall a_i \in s(t).$$

Therefore, by Corollary 11, for all $i \in [n]$ and all $l \in [k]$, we have

$$
\begin{aligned}
2B\sqrt{\alpha N_i(t_l)\log T} \geq & N_i(t_l)UCB_i(t_l) - \sum_{\tau=1}^{t_l} Q_{s(\tau)}(a_i) \\
\geq & N_i(t_l)\rho(t_l)\left(1 - \frac{1}{N_i(t)}\right) - \sum_{\tau=1}^{t_l} Q_{s(\tau)}(a_i) \\
\geq & N_i(t_l)Q_{s^*}(a_l) - \sum_{\tau=1}^{t_l} Q_{s(\tau)}(a_i) - Q_{s^*}(a_l),
\end{aligned}
\tag{1}
$$

where the last step follows from that $t_l$ is the last time step with $\rho(t_l) \geq Q_{s^*}(a_l)$. Summing up for $i \geq l+1$, we have

$$2B\sum_{i=l+1}^{n}\sqrt{\alpha N_i(t_l)\log T} \geq \sum_{i=l+1}^{n} Q_{s^*}(a_l)N_i(t_l) - \sum_{i=l+1}^{n}\sum_{\tau=1}^{t_l} Q_{s(\tau)}(a_i) - nQ_{s^*}(a_l).$$

Notice that for arms $a_i$ with $i \leq l$, we have $\sum_{i=1}^{l} Q_{s^*}(a_i)N_i(t_l) - \sum_{i=1}^{l}\sum_{\tau=1}^{t_l} Q_{s(\tau)}(a_i) \leq 0$ as $Q_{s(\tau)}(a_i) \geq Q_{s^*}(a_i)$ for all $\tau$ and $i \leq l$, by Assumption 1. Therefore we have

$$2B\sum_{i=l+1}^{n}\sqrt{\alpha N_i(t_l)\log T} \geq \sum_{i=1}^{l} Q_{s^*}(a_i)N_i(t_l) + \sum_{i>l}^{n} Q_{s^*}(a_l)N_i(t_l) - \sum_{i=1}^{n}\sum_{\tau=1}^{t_l} Q_{s(\tau)}(a_i) - nQ_{s^*}(a_l).$$

For the first two terms on the right side, we have $\sum_{i=1}^{l} Q_{s^*}(a_i)N_i(t_l) = \sum_{i=1}^{l} Q_{s^*}(a_i)t_l - \sum_{i=1}^{l} Q_{s^*}(a_i)(t_l - N_i(t_l))$ and $\sum_{i>l} Q_{s^*}(a_l)N_i(t_l) = Q_{s^*}(a_l)\left[(k-l)t_l + \sum_{i=1}^{l}(t_l - N_i(t_l))\right]$. Therefore, we have

$$
\begin{aligned}
2B\sum_{i=l+1}^{n}\sqrt{\alpha N_i(t_l)\log T} \geq & \sum_{i=1}^{l} Q_{s^*}(a_i)t_l + (k-l)Q_{s^*}(a_l)t_l - \sum_{i=1}^{n}\sum_{\tau=1}^{t_l} Q_{s(\tau)}(a_i) - \sum_{i=1}^{l-1}\delta_{il}(t_l - N_i(t_l)) - nQ_{s^*}(a_l) \\
\geq & \sum_{i=1}^{l} Q_{s^*}(a_i)t_l + (k-l)Q_{s^*}(a_l)t_l - \sum_{i=1}^{n}\sum_{\tau=1}^{t_l} Q_{s(\tau)}(a_i) - \sum_{i=1}^{l-1}\delta_{il}(t_i - t_i') - nQ_{s^*}(a_l).
\end{aligned}
$$

Recall that $t_i'$ is the number of optimal set played before $t_i$. Combined with Lemma 12, which states that $a_i$ is always played after $t_i$, the second inequality above follows from $t_l - N_i(t_l) = t_i - N_i(t_i) \leq t_i - t_i'$. The first inequality in Lemma 13 follows from

$$2B\sum_{i=l+1}^{n}\sqrt{\alpha N_i(t_l)\log T} \leq B\sqrt{4n\sum_{i=l+1}^{n}\alpha N_i(t_l)\log T} \leq B\sqrt{4\alpha kn\left(t_l - \frac{l}{k}t_l'\right)\log T}.$$

**Proof of the second inequality.** We reuse the following inequality that we proved at Equation (1), for all $i \in [n]$ and all $l \in [k]$, we have:

$$2B\sqrt{\alpha N_i(t_l)\log T} \geq N_i(t_l)Q_{s^*}(a_l) - \sum_{\tau=1}^{t_l} Q_{s(\tau)}(a_i) - Q_{s^*}(a_l).$$

Now, instead of summing over $i \geq l+1$, we sum over $i > k$ and have

$$\sum_{i=k+1}^{n} 2B\sqrt{\alpha N_i(t_l)\log T} \geq \sum_{i=k+1}^{n} N_i(t_l)Q_{s^*}(a_l) - \sum_{i=k+1}^{n}\sum_{\tau=1}^{t_l} Q_{s(\tau)}(a_i) - nQ_{s^*}(a_l).$$

Notice that for arms $a_i$ with $i \leq k$, we have $\sum_{i=1}^{k} Q_{s^*}(a_i)N_i(t_l) - \sum_{i=1}^{k}\sum_{\tau=1}^{t_l} Q_{s(\tau)}(a_i) \leq 0$ as $Q_{s(\tau)}(a_i) \geq Q_{s^*}(a_i)$ for all $\tau$ and $i \leq k$, by Assumption 1. Therefore we have

$$\sum_{i=k+1}^{n} 2B\sqrt{\alpha N_i(t_l)\log T} \geq \sum_{i=k+1}^{n} N_i(t_l)Q_{s^*}(a_l) + \sum_{i=1}^{k} Q_{s^*}(a_i)N_i(t_l) - \sum_{i=1}^{n}\sum_{\tau=1}^{t_l} Q_{s(\tau)}(a_i) - nQ_{s^*}(a_l).$$

The first inequality in Lemma 13 follows by

$$2B\sum_{i=k+1}^{n}\sqrt{\alpha N_i(t_l)\log T} \leq B\sqrt{4n\sum_{i=k+1}^{n}\alpha N_i(t_l)\log T} \leq B\sqrt{4\alpha kn(t_l - t_l')\log T}.$$

This completes the proof. $\qquad\square$

Recall that we assumed $a_1, \cdots, a_k$ all belong to $s^*$, with $Q_{s^*}(a_1) \geq Q_{s^*}(a_2) \geq \cdots \geq Q_{s^*}(a_k)$. Recall $\delta_{ij} = Q_{s^*}(a_i) - Q_{s^*}(a_j)$ and $\Delta_l = \sum_{i=l}^{k}\delta_{li}$.

**Lemma 14.** *Let* $\sigma_{ij} = \frac{4\delta_{ij}(\Delta_j+\epsilon)}{(\Delta_i+\epsilon)^2}$, *we have*

$$\sum_{j=i}^{k}\sigma_{ij} \leq 2, \ \forall i \leq k, \forall \epsilon \geq 0.$$

*Proof.* Expanding the summation, we have

$$\sum_{j=i}^{k}\sigma_{ij} = \sum_{j=i}^{k}\frac{4\delta_{ij}(\Delta_j+\epsilon)}{(\Delta_i+\epsilon)^2} = 4\sum_{j=i}^{k}\frac{\delta_{ij}}{\Delta_i+\epsilon}\left(\sum_{m=j}^{k}\frac{\delta_{jm}}{\Delta_i+\epsilon} + \frac{\epsilon}{\Delta_i+\epsilon}\right).$$

Note that

$$\sum_{m=j}^{k}\delta_{jm} + \sum_{m=i}^{j}\delta_{im} + \epsilon \leq \Delta_i + \epsilon \implies \sum_{m=j}^{k}\frac{\delta_{jm}}{\Delta_i+\epsilon} + \frac{\epsilon}{\Delta_i+\epsilon} \leq 1 - \sum_{m=i}^{j}\frac{\delta_{im}}{\Delta_i+\epsilon}.$$

For brevity, let $x_m = \frac{\delta_{im}}{\Delta_i+\epsilon}$, we have $\sum_{m=i}^{k}x_m \leq 1$ and

$$\sum_{j=i}^{k}\sigma_{ij} \leq 4\sum_{j=i}^{k}x_j\left(1 - \sum_{m=i}^{j}x_m\right) \leq 4\int_0^1 (1-x)dx \leq 2.$$

This completes the proof. $\qquad\square$

Follow the definition of $\sigma_{ij} = \frac{4\delta_{ij}(\Delta_j+\epsilon)}{(\Delta_i+\epsilon)^2}$ in Lemma 14. We have the following result.

**Lemma 15.** *For any* $1 \leq i < j \leq k$, *define function* $f(i,j) = 0.4\sigma_{ij} + \sum_{m=i+1}^{j-1} 0.4\sigma_{im}f(m,j)$, *we have*

1. $f(i,j) = 0.4\sigma_{ij} + \sum_{m=i+1}^{j-1} 0.4f(i,m)\sigma_{mj}, \quad \forall 1 \le i < j \le k$

2. $f(i,j) \le 1, \quad \forall 1 \le i < j \le k$

*Proof.* We first prove the first part. Let $\Pi(i,j)$ be the power set of $\{i, i+1, \cdots, j-1, j\}$. Let $\Gamma(i,j) = \{x | x \in \Pi(i,j), i \in x, j \in x\}$. Further, for $x \in \Gamma(i,j)$ defining

$$g(x) = \sigma_{x_1 x_2} \cdot \sigma_{x_2 x_3} \cdots \sigma_{x_{|x|-1} x_{|x|}}.$$

For example, for $x = \{2, 3, 5, 7\}$, we have $g(x) = \sigma_{23} \cdot \sigma_{35} \cdot \sigma_{57}$.

We first show by induction that $f(i,j) = \sum_{x \in \Gamma(i,j)} 0.4^{|x|-1} g(x)$. For base case $j = i+1$, we have $f(i,j) = 0.4\sigma_{ij}$. Now suppose $f(i,j) = \sum_{x \in \Gamma(i,j)} 0.4^{|x|-1} g(x)$ holds for $j - i \le c$, we proceed to prove it holds for $j - i = c + 1$.

Take $j = i + c + 1$, we have

$$
\begin{aligned}
f(i,j) &= 0.4\sigma_{ij} + \sum_{m=i+1}^{j-1} 0.4\sigma_{im} f(mj) \\
&= 0.4\sigma_{ij} + \sum_{m=i+1}^{j-1} \left( 0.4\sigma_{im} \sum_{x \in \Gamma(m,j)} 0.4^{|x|-1} g(x) \right) \\
&= 0.4\sigma_{ij} + \sum_{m=i+1}^{j-1} \sum_{x \in \Gamma(m,j)} 0.4^{|x|} \sigma_{im} g(x) \\
&= \sum_{x \in \Gamma(i,j)} 0.4^{|x|-1} g(x),
\end{aligned}
$$

where the last steps holds as a element $x$ belongs to $\Gamma(i,j)$ if and only if $x$ is one of the following two cases:

1. $x = \{i, j\}$,

2. $x = \{i, m, \cdots, j\}$ for some $m \in [i+1, j-1]$, and $\{m, \cdots, j\} \in \Gamma(m,j)$ by definition.

Therefore, we conclude via induction that

$$f(i,j) = \sum_{x \in \Gamma(i,j)} 0.4^{|x|-1} g(x). \tag{2}$$

Further, Equation (2) implies the first equation in Lemma 15, since a element $x$ belongs to $\Gamma(i,j)$ if and only if $x$ is one of the following two cases:

1. $x = \{i, j\}$,

2. $x = \{i, \cdots, m, j\}$ for some $m \in [i+1, j-1]$, and $\{i, \cdots, m\} \in \Gamma(i,m)$ by definition.

For the second part of the proof, we prove by induction. For the base case $j - i = 1$, we have $f(i,j) = 0.4\sigma_{ij} \le 0.4 \times 4 \times 0.5 \le 1$. Now, suppose that the inequality holds for any $i, j$ with $j - i = c$, then for any $i, j \le k$ with $j - i = c + 1$, we have

$$f(i,j) \le 0.4\sigma_{ij} + \sum_{m=i+1}^{j-1} 0.4\sigma_{im} = 0.4 \sum_{m=i+1}^{j} \sigma_{im} \le 0.4 \sum_{m=i}^{k} \sigma_{im} \le 0.8.$$

The last inequality follows from Lemma 14, which states $\sum_{m=i}^{k} \sigma_{im} \le 2$. $\square$

## D. Proof for Section 4

### D.1. Proof of Lemma 5

*Proof.* The rest of the proof conditions on the event that the inequality in Lemma 9 holds (and therefore the inequality in Corollary 10 holds), which happens with probability at least $1 - \frac{2}{T}$.

By Corollary 10, we have

$$UCB_i(t) \geq \frac{\sum_{\tau=1}^{t} Q_{s(\tau)}(a_i)}{N_i(t)},$$

for all $a_i$ and all $t \in [T]$. By Assumption 1, we have that $Q_s(a_i) \geq Q_{s^*}(a_i)$ for all $a_i \in s^*$ and all $s$ containing $a_i$. Therefore $\frac{\sum_{\tau=1}^{t} Q_{s(\tau)}(a_i)}{N_i(t)} \geq Q_{s^*}(a_i)$. This completes the proof that with probability at least $1 - \frac{2}{T}$, we have $UCB_i(t) \geq Q_{s^*}(a_i)$ simultaneously for all $a_i \in s^*$ and $t \in [T]$. $\square$

### D.2. Proof of Lemma 7

*Proof.* The rest of the proof conditions on the event that the inequality in Lemma 9 holds (and therefore the inequalities in Corollary 11 and lemmas 12 and 13 hold), which happens with probability at least $1 - \frac{2}{T}$.

Define $\delta_{ij} = Q_{s^*}(a_i) - Q_{s^*}(a_j)$, $\Delta_l = \sum_{i=l}^{k} \delta_{li}$. Define $t_l$ to be the last time step with $\rho(t_l) \geq Q_{s^*}(a_l)$. Denote $t'_l$ to be the number of times $s(t) = s^*$ for $t \leq t_l$. Note that if $\rho(T) \geq Q_{s^*}(a_l)$, then $t_l = T$ by definition.

**Case I:** $\Delta_l \geq \frac{\epsilon}{10}$.

By the first inequality of Lemma 13, we have

$$B\sqrt{4\alpha kn \left(t_l - \frac{l}{k}t'_l\right)\log T} \geq \sum_{i=1}^{l} Q_{s^*}(a_i)t_l + (k-l)Q_{s^*}(a_l)t_l - \sum_{i=1}^{n}\sum_{\tau=1}^{t_l} Q_{s(\tau)}(a_i) - \sum_{i=1}^{l-1} \delta_{il}(t_i - t'_i) - nQ_{s^*}(a_l).$$

Note that

$$\sum_{i=1}^{l} Q_{s^*}(a_i) + (k-l)Q_{s^*}(a_l) - \sum_{i=1}^{k} Q_{s^*}(a_i) = \sum_{i=l}^{k} \delta_{li} = \Delta_l.$$

By the fact $\sum_{i=1}^{k} Q_{s^*}(a_i) - \sum_{i=1}^{n} Q_{s(t)}(a_i) \geq \epsilon$ for all suboptimal set $s(t)$, we have

$$B\sqrt{4\alpha kn \left(t_l - \frac{l}{k}t'_l\right)\log T} \geq \Delta_l t_l + \epsilon(t_l - t'_l) - \sum_{i=1}^{l-1} \delta_{il}(t_i - t'_i) - nQ_{s^*}(a_l)$$

$$\geq (\Delta_l + \epsilon)\left(t_l - \frac{l}{k}t'_l\right) - \frac{\epsilon(k-l) - \Delta_l l}{k}t'_l - \sum_{i=1}^{l-1} \delta_{il}(t_i - t'_i) - nQ_{s^*}(a_l). \quad (3)$$

Next, we prove by mathematical induction that $(t_i - t'_i) \leq \left(3.08 + 10\sum_{j=1}^{i-1} f(j,i)\right)\frac{4\alpha B^2 kn \log T}{(\Delta_i + \epsilon)^2}$, where the function $f(i,j)$ is defined in Lemma 15. For notation simplicity, we write $t_i - t'_i$ in the following form

$$t_i - t'_i \leq c_i \frac{4\alpha B^2 kn \log T}{(\Delta_i + \epsilon)^2},$$

and proceed to bound $c_i$. With the new convention, we can rewrite Equation (3) as

$$B\sqrt{4\alpha kn \left(t_l - \frac{l}{k}t'_l\right)\log T} \geq (\Delta_l + \epsilon)\left(t_l - \frac{l}{k}t'_l\right) - \frac{\epsilon(k-l) - \Delta_l l}{k}t'_l - \sum_{i=1}^{l-1} \delta_{il}c_i \frac{4\alpha B^2 kn \log T}{(\Delta_i + \epsilon)^2} - nQ_{s^*}(a_l). \quad (4)$$

For $\log T \geq 2.5$, with the fact $k \geq \Delta_l + \epsilon, 1 \geq Q_{s^*}(a_l), \alpha \geq 2$, we have $4n(\Delta_l + \epsilon)Q_{s^*}(a_l) \leq 0.8\alpha kn \log T$ and therefore solving Equation (4) for the bound for $\sqrt{t_l - \frac{l}{k}t'_l}$, we have

$$\sqrt{t_l - \frac{l}{k}t'_l} \leq \frac{1}{2}\left(1 + \sqrt{1.2 + \sum_{i=1}^{l-1}\frac{4\delta_{il}(\Delta_l + \epsilon)}{(\Delta_i + \epsilon)^2}c_i + \frac{4(\Delta_l + \epsilon)\frac{\epsilon(k-l)-\Delta_l l}{k}}{4\alpha B^2 kn \log T}t'_l}\right)\frac{B\sqrt{4\alpha kn \log T}}{\Delta_l + \epsilon}.$$

Further, define $\sigma_{il} = \frac{4\delta_{il}(\Delta_l + \epsilon)}{(\Delta_i + \epsilon)^2}$, we have

$$\sqrt{t_l - \frac{l}{k}t'_l} \leq \frac{1}{2}\left(1 + \sqrt{1.2 + \sum_{i=1}^{l-1}\sigma_{il}c_i + \frac{4(\Delta_l + \epsilon)\frac{\epsilon(k-l)-\Delta_l l}{k}}{4\alpha B^2 kn \log T}t'_l}\right)\frac{B\sqrt{4\alpha kn \log T}}{\Delta_l + \epsilon}.$$

By the fact $(1 + a)^2 \leq 1.1a^2 + 11$ for any real number $a$, we have

$$t_l - \frac{l}{k}t'_l \leq \frac{1}{4}\left(11 + 1.32 + 1.1\sum_{i=1}^{l-1}\sigma_{il}c_i\right)\frac{4\alpha B^2 kn \log T}{(\Delta_l + \epsilon)^2} + 1.1\frac{\epsilon(k-l) - \Delta_l l}{k(\Delta_l + \epsilon)}t'_l.$$

Since $\Delta_l \geq \frac{\epsilon}{10}$, we have $\frac{1.1\epsilon k - 1.1(\Delta_l + \epsilon)l}{(\Delta_l + \epsilon)k} \leq \frac{(\Delta_l + \epsilon)k - (\Delta_l + \epsilon)l}{(\Delta_l + \epsilon)k} = \frac{k-l}{k}$. Therefore we have

$$t_l - \frac{l}{k}t'_l \leq \left(3.08 + 0.275\sum_{i=1}^{l-1}\sigma_{il}c_i\right)\frac{4\alpha B^2 kn \log T}{(\Delta_l + \epsilon)^2} + \frac{k-l}{k}t'_l$$

$$\implies t_l - t'_l \leq \left(3.08 + 0.275\sum_{i=1}^{l-1}\sigma_{il}c_i\right)\frac{4\alpha B^2 kn \log T}{(\Delta_l + \epsilon)^2}.$$

Plug in the convention of $c_l$, we have

$$c_l \leq 3.08 + 0.275\sum_{i=1}^{l-1}\sigma_{il}c_i.$$

Now we proceed to show $c_i \leq 3.08 + 10\sum_{j=1}^{i-1}f(j, i)$ by induction. For base case $i = 1, 2$, we have

$$c_1 \leq 3.08, \quad c_2 \leq 3.08 + 0.275\sigma_{12}c_1 \leq 3.08 + \sigma_{12} \leq 0.308 + 10f(1, 2).$$

Next, suppose $c_i \leq 3.08 + 10\sum_{j=1}^{i-1}f(j, i)$ holds for all $i \leq l - 1$, for $i = l$ we have

$$c_l \leq 3.08 + 0.275\sum_{i=1}^{l-1}\sigma_{il}c_i \leq 3.08 + 0.275\sum_{i=1}^{l-1}\left[3.08 + 10\sum_{j=1}^{i-1}f(j, i)\right]\sigma_{il}$$

$$\leq 3.08 + \sum_{i=1}^{l-1}\left[2.75\sigma_{il} + 2.75\sum_{j=1}^{i-1}f(j, i)\sigma_{il}\right]$$

$$\leq 3.08 + \sum_{j=1}^{l-1}\sum_{i=j+1}^{l-1}2.75f(j, i)\sigma_{il} + \sum_{i=1}^{l-1}2.75\sigma_{il}$$

$$\leq 3.08 + \sum_{j=1}^{l-1}\left[2.75\sigma_{jl} + \sum_{i=j+1}^{l-1}2.75f(j, i)\sigma_{il}\right]$$

$$\leq 3.08 + 10\sum_{j=1}^{l-1}f(j, l).$$

The last inequality follows from the first equation in Lemma 15: $f(i,j) = 0.4\sigma_{ij} + \sum_{m=i+1}^{j-1} 0.4 f(i,m)\sigma_{mj}$. This completes induction.

Combining with the second inequality in Lemma 15 which shows that $f(i,l) \leq 1, \forall i < l \leq k$, we have $c_l \leq 10l$. It therefore implies that $t_l - t'_l \leq \frac{40\alpha B^2 lkn \log T}{(\Delta_l + \epsilon)^2}$. This completes the proof of the first case in Lemma 7.

**Case II:** $\Delta_l < \frac{\epsilon}{10}$.

Denote $l'$ to be the largest $i$ with $\Delta_i \geq \epsilon/10$, and let $l' = 0$ if $\Delta_i < \epsilon/10$ for all $i \in [k]$. By definition, we know $l > l'$. Applying the second inequality in Lemma 13, we have that

$$
\begin{aligned}
2B\sqrt{\alpha kn (t_l - t'_l) \log T} &\geq \sum_{i=k+1}^{n} Q_{s^*}(a_l) N_i(t_l) + \sum_{i=1}^{k} Q_{s^*}(a_i) N_i(t_l) - \sum_{i=1}^{n} \sum_{\tau=1}^{t_l} Q_{s(\tau)}(a_i) - n Q_{s^*}(a_l) \\
&\geq \sum_{i=1}^{k} Q_{s^*}(a_i) t_l - \sum_{i=1}^{l-1} \left( Q_{s^*}(a_i) - Q_{s^*}(a_l) \right) (t_l - N_i(t_l)) - \sum_{i=1}^{n} \sum_{\tau=1}^{t_l} Q_{s(\tau)}(a_i) - n Q_{s^*}(a_l) \\
&\geq \epsilon (t_l - t'_l) - \sum_{i=1}^{l-1} \delta_{il}(t_l - N_i(t_l)) - n Q_{s^*}(a_l) \\
&\geq \epsilon (t_l - t'_l) - \sum_{i=1}^{l-1} \delta_{il}(t_i - t'_i) - n Q_{s^*}(a_l).
\end{aligned}
\tag{5}
$$

Recall that $t'_i$ is the number of optimal set played before $t_i$. Combined with Lemma 12, which states that $a_i$ is always played after $t_i$, the last inequality above follows from $t_l - N_i(t_l) = t_i - N_i(t_i) \leq t_i - t'_i$.

Next, we prove by mathematical induction that $(t_i - t'_i) \leq \left( 3.08 + 10 \sum_{j=1}^{i-1} f(j,i) \right) \frac{4\alpha B^2 kn \log T}{\epsilon^2}$ for all $i > l'$, where the function $f(i,j)$ is defined in Lemma 15. For notation simplicity, we write $t_i - t'_i$ for $i > l'$ in the following form

$$
t_i - t'_i \leq d_i \frac{4\alpha B^2 kn \log T}{\epsilon^2},
$$

and proceed to bound $d_i$. With this convention, we can rewrite Equation (5) as

$$
2B\sqrt{\alpha kn (t_l - t'_l) \log T} \geq \epsilon (t_l - t'_l) - \sum_{i=1}^{l'} \delta_{il} c_i \frac{4\alpha B^2 kn \log T}{(\Delta_i + \epsilon)^2} - \sum_{i=l'+1}^{l-1} \delta_{il} d_i \frac{4\alpha B^2 kn \log T}{\epsilon^2} - n Q_{s^*}(a_l)
$$

For $\log T \geq 2.5$, with the fact $k \geq \Delta_l + \epsilon, 1 \geq Q_{s^*}(a_l), \alpha \geq 2$, we have $4n(\Delta_l + \epsilon) Q_{s^*}(a_l) \leq 0.8\alpha kn \log T$ and therefore solving Equation (5) for the bound for $\sqrt{t_l - t'_l}$, we have

$$
\begin{aligned}
\sqrt{t_l - t'_l} &\leq \frac{B\sqrt{4\alpha kn \log T}\left( 1 + \sqrt{1.2 + \sum_{i=1}^{l'} \frac{4\delta_{il}\epsilon}{(\Delta_i + \epsilon)^2} c_i + \sum_{i=l'+1}^{l-1} \frac{4\delta_{il}\epsilon}{\epsilon^2} d_i} \right)}{2\epsilon} \\
&\leq \frac{B\sqrt{4\alpha kn \log T}\left( 1 + \sqrt{1.2 + \sum_{i=1}^{l'} \frac{4\delta_{il}(\Delta_l + \epsilon)}{(\Delta_i + \epsilon)^2} c_i + 1.21 \sum_{i=l'+1}^{l-1} \frac{4\delta_{il}(\Delta_l + \epsilon)}{(\Delta_i + \epsilon)^2} d_i} \right)}{2\epsilon}.
\end{aligned}
$$

The second inequality follows from $\frac{(\Delta_i + \epsilon)^2}{\epsilon^2} \leq 1.21$ as $\Delta_i < \frac{\epsilon}{10}$ for all $i > l'$. Define $\sigma_{il} = \frac{4\delta_{il}(\Delta_l + \epsilon)}{(\Delta_i + \epsilon)^2}$, with the convention of $d_l$, we have

$$
\sqrt{d_l} \leq \frac{1}{2}\left( 1 + \sqrt{1.2 + \sum_{i=1}^{l'} \sigma_{il} c_i + 1.21 \sum_{i=l'+1}^{l-1} \sigma_{il} d_i} \right).
$$

Again use the fact that $(1+a)^2 \le 11 + 1.1a^2$, we have

$$d_l \le \frac{1}{4}\left(11 + 1.32 + 1.1\sum_{i=1}^{l'}\sigma_{il}c_i + 1.331\sum_{i=l'+1}^{l-1}\sigma_{il}d_i\right)$$

$$\le 3.08 + 0.275\sum_{i=1}^{l'}\sigma_{il}c_i + 0.34\sum_{i=l'+1}^{l-1}\sigma_{il}d_i.$$

We next prove by induction that $d_i \le 3.08 + 10\sum_{j=1}^{i-1}f(j,i)$ for all $i > l'$. For the base case, $i = l' + 1$, we immediately have $d_i \le 3.08 + 0.275\sum_{j=1}^{i-1}\sigma_{ji}c_j = 3.08 + 10\sum_{j=1}^{i-1}f(j,i)$, which follows from the proof in **Case I**.

Next, assuming that $d_i \le 3.08 + 10\sum_{j=1}^{i-1}f(j,i)$ holds for $i \le l-1$, then for $i = l$ we have

$$d_l \le 3.08 + 0.275\sum_{i=1}^{l'}\sigma_{il}c_i + 0.34\sum_{i=l'+1}^{l-1}\sigma_{il}d_i$$

$$\le 3.08 + 0.275\sum_{i=1}^{l'}\left[3.08 + 10\sum_{j=1}^{i-1}f(j,i)\right]\sigma_{il} + 0.34\sum_{i=l'+1}^{l-1}\left[3.08 + 10\sum_{j=1}^{i-1}f(j,i)\right]\sigma_{il}$$

$$\le 3.08 + \sum_{i=1}^{l-1}\left[3.4\sigma_{il} + 3.4\sum_{j=1}^{i-1}f(j,i)\sigma_{il}\right]$$

$$\le 3.08 + \sum_{j=1}^{l-1}\sum_{i=j+1}^{l-1}3.4f(j,i)\sigma_{il} + \sum_{i=1}^{l-1}3.4\sigma_{il}$$

$$\le 3.08 + \sum_{j=1}^{l-1}\left[3.4\sigma_{jl} + \sum_{i=j+1}^{l-1}3.4f(j,i)\sigma_{il}\right]$$

$$\le 3.08 + 10\sum_{j=1}^{l-1}f(j,l).$$

The last inequality follows from the first equation in Lemma 15: $f(i,j) = 0.4\sigma_{ij} + \sum_{m=i+1}^{j-1}0.4f(i,m)\sigma_{mj}$. This completes induction.

Combining with the second inequality in Lemma 15 which shows that $f(i,l) \le 1, \forall i < l \le k$, we have $d_l \le 10l$. It therefore implies that $t_l - t'_l \le \frac{40\alpha B^2 lkn\log T}{\epsilon^2}$. This completes the proof of the second case in Lemma 7. □

### D.3. Proof of Lemma 8

*Proof.* The rest of the proof conditions on the event that the inequality in Lemma 9 holds (and therefore inequalities in Lemmas 5 and 13 hold), which happens with probability at least $1 - \frac{2}{T}$.

Recall that for $a_l \in s^*$, $t_l$ is the last time step with $\rho(t_l) \ge Q_{s^*}(a_l)$ and $t'_l$ is the number of times $s(t) = s^*$ for $t \le t_l$. From Lemma 5, we know that the arms $a_i \in s^*$ all have $UCB_i(t) \ge Q_{s^*}(a_k)$ for all $t \in [T]$. Therefore, we have $\rho'(t) \ge Q_{s^*}(a_k)$ for all $t$ and thus $\rho(T) \ge Q_{s^*}(a_k)$. It implies that $t_k = T$ and $R(T) = R(t_k)$. Plug in the first inequality in Lemma 13 with $l = k$, we have

$$B\sqrt{4\alpha kn(t_k - t'_k)\log T} \ge \sum_{i=1}^{k}Q_{s^*}(a_i)t_k - \sum_{i=1}^{n}\sum_{\tau=1}^{t_k}Q_{s(\tau)}(a_i) - \sum_{i=1}^{k-1}\delta_{ik}(t_i - t'_i) - nQ_{s^*}(a_k).$$

Note that $R(t_k) = \sum_{i=1}^{k} Q_{s^*}(a_i) t_k - \sum_{i=1}^{n} \sum_{\tau=1}^{t_k} Q_{s(\tau)}(a_i)$, rearranging the terms, we have

$$R(T) = R(t_k) \leq B\sqrt{4\alpha k n (t_k - t_k') \log T} + \sum_{l=1}^{k-1} \delta_{lk} (t_l - t_l') + n Q_{s^*}(a_k).$$

This completes the proof. $\qquad\qquad\qquad\qquad\qquad\qquad\qquad\qquad\qquad\qquad\qquad\qquad\qquad\qquad$ □

### D.4. Proof of Theorem 3

Now we are ready to prove Theorem 3.

*Proof.* We first consider when the inequality in Lemma 9 does not hold for some $t$ and arm $a_i$. Since this happens with probability at most $2/T$ and the regret is trivially bounded by $T$. Therefore it induces at most 2 regret in expectation.

The rest of the proof conditions on the event that the inequality in Lemma 9 holds (and therefore the inequalities in Lemmas 7 and 8 hold), which happens with probability at least $1 - \frac{2}{T}$.

We first prove the gap-dependent regret bound. Combining Lemmas 7 and 8, we have

$$\begin{aligned}
R(T) \leq &\; 2B\sqrt{\alpha k n (t_k - t_k') \log T} + \sum_{l=1}^{k-1} \delta_{lk} (t_l - t_l') + n Q_{s^*}(a_k) \\
\leq &\; \frac{14\alpha B^2 k^{\frac{3}{2}} n \log T}{\epsilon} + \sum_{l=1}^{k-1} \delta_{lk} (t_l - t_l') + n Q_{s^*}(a_k) \\
\overset{(a)}{\leq} &\; \frac{14\alpha B^2 k^{\frac{3}{2}} n \log T}{\epsilon} + \sum_{l=1}^{k-1} \frac{40 \alpha B^2 l k n \log T}{\epsilon} + n \\
\leq &\; \frac{35 \alpha B^2 k^3 n \log T}{\epsilon}.
\end{aligned}$$

Recall that $\delta_{ik} = Q_{s^*}(a_i) - Q_{s^*}(a_k)$, and $\Delta_l = \sum_{i=l}^{k} \delta_{li}$. The inequality (a) used the fact that

$$\delta_{lk} \cdot (t_l - t_l') \leq \delta_{lk} \cdot \frac{40 \alpha B^2 l k n \log T}{(\Delta_l + \epsilon)^2} \leq \frac{40 \alpha B^2 l k n \log T}{\epsilon}, \quad \forall \Delta_l \geq \epsilon/10;\; and$$

$$\delta_{lk} \cdot (t_l - t_l') \leq \delta_{lk} \cdot \frac{40 \alpha B^2 l k n \log T}{\epsilon^2} \leq \frac{40 \alpha B^2 l k n \log T}{\epsilon}, \quad \forall \Delta_l < \epsilon/10.$$

This completes the proof of the gap-dependent regret bound $O\left(\frac{B^2 k^3 n \log T}{\epsilon}\right)$.

For the gap-independent part, recall that $\delta_{ik} = Q_{s^*}(a_i) - Q_{s^*}(a_k)$, and $\Delta_l = \sum_{i=l}^{k} \delta_{li}$.

If $\Delta_1 + \epsilon < 10kB\sqrt{\frac{\alpha n \log T}{T}}$, from Lemma 8, we have

$$\begin{aligned}
R(T) \leq &\; 2B\sqrt{\alpha k n T \log T} + \sum_{i=1}^{k} \delta_{ik} T + n \\
\leq &\; 2B\sqrt{\alpha k n T \log T} + \sum_{i=1}^{k} \Delta_1 T + n = O\left(Bk^2\sqrt{nT\log T}\right),
\end{aligned}$$

where we used the fact that $t_l - t_l' \leq t_l \leq T$ for all $l \in [k]$, and $\delta_{ik} \leq \Delta_1$ for all $i \in [k]$.

On the other hand, if $\Delta_1 + \epsilon < 10kB\sqrt{\frac{\alpha n \log T}{T}}$, let $m$ denote the largest $i \in [1, k]$ such that $\Delta_i + \epsilon \geq 10kB\sqrt{\frac{\alpha n \log T}{T}}$. To invoke Lemma 7, we need to discuss the relationship between $\Delta_m$ and $\epsilon/10$

**(a)** If $\Delta_m \geq \epsilon/10$, combining Lemmas 7 and 8, we have

$$R(T) \leq 2B\sqrt{\alpha k n T \log T} + \sum_{l=1}^{m} \frac{40\alpha B^2 lkn \log T}{\Delta_m + \epsilon} + \sum_{i=m+1}^{k} \delta_{ik} T + n$$

$$\leq 2B\sqrt{\alpha k n T \log T} + \frac{20\alpha B^2 k^3 n \log T}{\Delta_m + \epsilon} + \sum_{i=m+1}^{k} \delta_{ik} T + n.$$

If $m = k$, we do not have the third term. Otherwise, by definition of $\Delta_{m+1}$, we have $\delta_{ik} \leq \Delta_{m+1}, \forall i \geq m+1$. Therefore, we have

$$R(t) \leq 2B\sqrt{\alpha k n T \log T} + \frac{20\alpha B^2 k^3 n \log T}{\Delta_m + \epsilon} + (k-m)\Delta_{m+1}T + n.$$

With $\Delta_m + \epsilon \geq 10kB\sqrt{\frac{\alpha n \log T}{T}} \geq \Delta_{m+1} + \epsilon$ and the fact that $T > n$, we have the $O(Bk^2\sqrt{nT \log T})$ gap-independent regret bound.

**(b)** If $\Delta_m < \epsilon/10$, $\Delta_m + \epsilon \geq 10kB\sqrt{\frac{\alpha n \log T}{T}}$ implies that $\epsilon \geq 9kB\sqrt{\frac{\alpha n \log T}{T}}$. Combining Lemmas 7 and 8, we have

$$R(T) \leq 2B\sqrt{\alpha k n T \log T} + \sum_{l=1}^{m} \frac{40\alpha B^2 lkn \log T}{\epsilon} + \sum_{i=m+1}^{k} \delta_{ik} T + n$$

$$\leq 2B\sqrt{\alpha k n T \log T} + \frac{20\alpha B^2 k^3 n \log T}{\epsilon} + \sum_{i=m+1}^{k} \delta_{ik} T + n.$$

If $m = k$, we do not have the third term. Otherwise, by definition of $\Delta_{m+1}$, we have $\delta_{ik} \leq \Delta_{m+1}, \forall i \geq m+1$. Therefore, we have

$$R(t) \leq 2B\sqrt{\alpha k n T \log T} + \frac{20\alpha B^2 k^3 n \log T}{\epsilon} + (k-m)\Delta_{m+1}T + n.$$

With $\epsilon \geq 9kB\sqrt{\frac{\alpha n \log T}{T}}$ and $10kB\sqrt{\frac{\alpha n \log T}{T}} \geq \Delta_{m+1} + \epsilon$ and the fact that $T > n$, we have the $O(Bk^2\sqrt{nT \log T})$ gap-independent regret bound.

Combining all the cases completes the proof. $\square$

# E. Experiment Setup

## E.1. Multinomial Logit

In this environment, the reward is generated according to a multinomial logit model

$$Q_{s(t)}(a_i) = \frac{v_i}{1 + \sum_{a_i \in s(t)} v_i}, \quad Q_{s(t)}(a_0) = \frac{1}{1 + \sum_{a_i \in s(t)} v_i}.$$

where $v_i$ is the value associated with each arm $a_i$, determining the reward probability. In this experiment, we set $v_i = 1 - 0.04i$ with $i \in [20]$. The size of set is set to $k = 10$, and the optimal set is $s^*$ is composed by arms from $a_1$ to $a_{10}$. The regret of set $s(t)$ is given by

$$reg(s(t)) = \frac{1}{1 + \sum_{a_i \in s(t)} v_i} - \frac{1}{1 + \sum_{a_i \in s^*} v_i}.$$

## E.2. Random Utility Model

In this environment, for an set $s(t)$ at time step $t$, each arm $a_i \in s(t)$ will independently draw a Gaussian distributed random variable $U_i \sim \mathcal{N}(\mu_i, 1)$, where $\mu_i$ is the mean associated with each arm $a_i$. Along with that $a_0$ will draw a $U_0 \sim \mathcal{N}(2, 1)$. The arm $a_i$ (including $a_0$) with highest $U_i$ will receive reward. Thus we have the probability of $a_i$ getting reward as

$$Q_{s(t)}(a_i) = \mathbb{P}(U_i = \max_{a_j \in s(t) \cup \{a_0\}} U_j).$$

Here, we set $\mu_i = 1 - 0.04i$ with $i \in [20]$. The size of set is set to $k = 5$, and the optimal set $s^*$ is composed by the arms from $a_1$ to $a_5$. For the convenience of computation, the regret of set $s(t)$ is defined slightly different as

$$reg(s(t)) = \sum_{a_i \in s^*} \mu_i - \sum_{a_i \in s(t)} \mu_i.$$

Once $s(t)$ recovers the optimal set $s^*$, which maximizes the probability of $s(t)$ receiving reward, we will have this regret $reg(s(t)) = 0$.

### E.3. Preference Matrix

In this environment, the probability of one arm $a_i$ getting reward is fully specified by a preference matrix. For ease of representation, we set the number of arms to $n = 10$ and the size of set to $k = 2$. Th total number of sets is 45, much lesser than the previous two environments. However, with a specially designed preference matrix (including the loop in preference, etc), the environment turns out to be the hardest.

We set $M$ to be the preference matrix with $M_{i,j} = Q_{\{a_i,a_j\}}(a_i) - Q_{\{a_i,a_j\}}(a_j)$. We set the optimal set to be $s^* = \{a_1, a_2\}$ with $Q_{\{a_1,a_2\}}(a_1) + Q_{\{a_1,a_2\}}(a_2) = 0.92$. For all other sets $s$ which are sub-optimal, we set $Q_{\{a_i,a_j\}}(a_i) + Q_{\{a_i,a_j\}}(a_j) = 0.9$. The preference matrix $M$ is given in Table 2.

|        | $a_1$  | $a_2$  | $a_3$  | $a_4$  | $a_5$  | $a_6$  | $a_7$  | $a_8$  | $a_9$  | $a_{10}$ |
|--------|--------|--------|--------|--------|--------|--------|--------|--------|--------|----------|
| $a_1$    | –      | 0.02   | 0.05   | 0.1    | 0.1    | 0.2    | 0.25   | 0.3    | 0.3    | 0.3      |
| $a_2$    | -0.02  | –      | 0.05   | 0.1    | 0.1    | 0.2    | 0.25   | 0.3    | 0.3    | 0.3      |
| $a_3$    | -0.05  | -0.05  | –      | 0.45   | 0.45   | 0.45   | 0.45   | 0.45   | 0.45   | 0.45     |
| $a_4$    | -0.1   | -0.1   | -0.45  | –      | -0.3   | 0.3    | 0      | 0      | 0      | 0        |
| $a_5$    | -0.1   | -0.1   | -0.45  | 0.3    | –      | -0.3   | 0      | 0      | 0      | 0        |
| $a_6$    | -0.2   | -0.2   | -0.45  | -0.3   | 0.3    | –      | 0      | 0      | 0      | 0        |
| $a_7$    | -0.25  | -0.25  | -0.45  | 0      | 0      | 0      | –      | 0      | 0      | 0        |
| $a_8$    | -0.3   | -0.3   | -0.45  | 0      | 0      | 0      | 0      | –      | 0      | 0        |
| $a_9$    | -0.3   | -0.3   | -0.45  | 0      | 0      | 0      | 0      | 0      | –      | 0        |
| $a_{10}$ | -0.3   | -0.3   | -0.45  | 0      | 0      | 0      | 0      | 0      | 0      | –        |

*Table 2.* Preference Matrix $M$

We can see that when $a_3$ pairs with any other sub-optimal arm, it will have a higher chance of getting reward than $a_1$ and $a_2$. It makes $a_3$ the seemingly best single arm. Also note that when $a_4$ pairs with $a_5$, $a_5$ will have a higher chance of getting reward. Similarly, $a_6$ will win over $a_5$ and $a_4$ will win over $a_6$. The preference therefore forms a loop among $a_4, a_5, a_6$.

The regret of $\{a_i, a_j\}$ is given by

$$reg(\{a_i, a_j\}) = Q_{\{a_1,a_2\}}(a_1) + Q_{\{a_1,a_2\}}(a_2) - Q_{\{a_i,a_j\}}(a_i) - Q_{\{a_i,a_j\}}(a_j).$$

### E.4. Random Weak Optimal Set Consistency

In this environment, we randomly generate the environment with Algorithm 2 that satisfies the Assumption 1.

By construction, the environment satisfies Assumption 1. Moreover, as we randomly sample the feedback for each set randomly, it's not necessary for the generated environment to satisfy more stronger Assumption, e.g. the strict preference order. The regret of set $s(t)$ is given by

$$reg(s(t)) = \sum_{a \in s(t)} Q_{s(t)}(a) - \sum_{a^* \in s^*} Q_{s^*}(a^*).$$

---

**Algorithm 2** GENERATING ENVIRONMENT SATISFIES ASSUMPTION 1.

---

1: **Input:** Number of Arms $n$, set Size $k$.
2: Set set $s^* = \{1, 2, \cdots, k\}$ be the optimal set. Randomly Sample $Q_{s^*}(a) \sim \text{Uniform}(0, \frac{1}{k})$. The rewards are binary reward, with expectation generated as following:
3: **for** set $s \neq s^*$ **do**
4:    **while** $\sum_{a \in s} Q_s(a) > \sum_{a^* \in s^*} Q_{s^*}(a^*)$ **do**
5:       **for** $a \in s$ **do**
6:          **if** $a \in s^*$ **then**
7:             Sample $Q_s(a) \sim \text{Uniform}(Q_{s^*}(a), \frac{1}{k})$.
8:          **else**
9:             Sample $Q_s(a) \sim \text{Uniform}(0, \frac{1}{k})$.
10:          **end if**
11:       **end for**
12:    **end while**
13: **end for**

---

