# OpenReview forum: "UCB Provably Learns From Inconsistent Human Feedback"
_ICML.cc/2023/Workshop/ILHF — ILHF Workshop ICML 2023_

### Official Review · Reviewer_vFWv · 2023-06-13
**Solid paper proposing + solving new variant of combinatorial bandits**

**Rating:** 7
**Confidence:** 3

**Review:**


**Two sentence summary**: Authors consider the performance of UCB algorithms in the combinatorial bandits setting where the preference over arms can flip between different sets. They propose a new consistency criteria for preferences that generalizes existing consistency assumptions, and prove that a simple UCB algorithm achieves regret close to the lower bound in their setup.

**Summary and contributions**
Authors seek to answer the question -- why does UCB seem to work well in practice for combinatorial bandits with semi-bandit feedback, even when some of the assumptions of the UCB-based algorithms are not justified? To this, they consider the problem of learning the optimal choice set when the value of each arm is _set-dependent_ (though the value of the choice set is still the sum of the values of its constituent arms). In this setting, the preferences over arms can be possibly consistency -- that is, the value of arm 1 could be higher than arm 2 in set 1 while arm 2 has higher value than arm 1 in set 2.

They propose a weaker consistency criterion in this setting -- _weak optimal set consistency_ (_WOSC_) -- which states that the value of the each of the arms in the optimal choice set is no higher than its value in any other set. They prove that with this assumption, a simple UCB algorithm is able to achieve $O\left ( \min(\frac{k^3n\log T}{\epsilon}, k^2 \sqrt{n T \log T} ) \right )$ regret, which is close to the $\Omega(\frac{n \log T}{\epsilon})$ regret lower bound they prove for the WOSC + combinatorial bandit setting. (Interestingly, the construction of the family of environments in the proof of the regret lower bound does not actually use set-dependent rewards!) Finally, they validate their results on experiments on 4 different toy bandit environments.

**Strengths and weaknesses**
* I think the authors do a good job of posing a puzzle -- if UCB works in practice despite inconsistent human preferences, can we figure out a weaker criteria that explains why it works?
* I found the motivating example helpful for understanding what the setting was, though I think the exposition could be clearer by situating the scenario with more realistic details -- it took me a bit to understand the value of `Digital Camera`.
* The authors did a good job of situating their WOSC criterion and explaining how previous models for combinatorial bandits all satisfy this criterion.
* I found their intuition in section 4.2 (esp their motivating example) and proof sketch in section 4.3 quite clear and helpful for understanding their proof of the main theorem in appendix D.
* I found the proof of the regret lower bound somewhat hard to follow, though this may be my lack of familarity with standard results in the literature. I think a proof sketch or intuition for theorem 4 would've helped me a lot in understanding the problem.
* Similarly, I found the actual proofs in appendices C and D to be significantly less clear than high level descriptions in the main body. I think including more description of the high level approaches (as well as restating the theorems/lemmas in the appendices) would've helped me follow along better.


**Explanation of ratings**
I'm not super familiar with the recent combinatorial bandit literature, so I'm unable to make confident judgments as to the claimed novelty of the analysis of the main UCB-based algoirthm or technical contributions of the results. That being said, I think the results are technically solid and the WOSC criterion proposed seems quite interesting as an alternative consistency criterion as opposed to the normal, stronger consistency criteria. So I think this is a solid accept for the workshop, though with only medium confidence.

Also note that while I did checked the correctness of proofs in appendix A, B, and C, I did not check the correctness of proofs in appendix D, including the proof of their main regret bound theorem). I'm also not super familiar with the UCB literature and may have missed errors.

---

### Official Review · Reviewer_9xZM · 2023-06-16
**Very clear and well written theory paper with relevant set up. Technical part still needs to be checked**

**Rating:** 9
**Confidence:** 2

**Review:**

Summary
The author studies and provides a bound for UCB algorithms for combinatorial bandits where there are set-dependent and inconsistent preferences under WOSC assumptions. The authors do this by illustrating they assumption with examples, showing its generality, and analyzing UCB algorithms

Pros:
    1. The authors correlates and contrasts the scope of their work very well from previous work repeatedly throughout the work, which strongly improves clarity and readability. The use of motivating example and proof sketch also helps break down the theoretical components.
    2. The problem setting of inconsistent preferences is very relevant to machine learning and recommendation systems in general.
    3. The results are backed by synthetic experiments in a easy-to-follow manner.

I am unable to check the quality of the theoretical proofs, and recommend further reviewing the soundness and whether there is any error in them.

---

### Decision · Program_Chairs · 2023-06-20

Accept